# Sensitivity study on the main tidal constituents of the Gulf of Tonkin by using the frequency-domain tidal solver in T-UGOm.

**Violaine Piton[1,2], Marine Herrmann [1,2], Florent Lyard[1], Patrick Marsaleix[(3)] , Thomas Duhaut[3], Damien Allain[1], Sylvain Ouillon[1,2]**

[(1)] LEGOS, IRD, UMR556 IRD/CNES/CNRS/University of Toulouse, 31400 Toulouse,
10 France
[(2)] LOTUS Laboratory, University of Science and Technology of Hanoi (USTH), Vietnam Academy of Science and Technology (VAST), 18 Hoang Quoc Viet, Cau Giay, Hanoi, Vietnam
[(3)] LA, CNRS, University of Toulouse, 31400 Toulouse, France
15 *Correspondence to:* Violaine Piton violaine.piton@legos.obs-mip.fr

**Abstract**

Tidal dynamics consequences on hydro-sedimentary processes are a recurrent issue in estuarine and coastal processes studies and accurate tidal solutions are a prerequisite for modelling sediment transport, especially in macro-tidal regions. The motivation for the study presented in this publication is to implement and optimize the model configuration that will satisfy this prerequisite in the frame of a larger objective to study the sediment dynamics and fate in the Red River delta to the Gulf of Tonkin from a numerical hydrodynamical-sediment coupled model. Therefore, we focus on the main tidal constituents to conduct sensitivity experiments on the bathymetry and bottom friction parameterization. The frequency-domain solver available in the hydrodynamic unstructured grid model T-UGOm has been used to reduce the computational cost and allow for wider parameter explorations. Tidal solutions obtained from the optimal configuration were evaluated from tide measurements derived from satellite altimetry and tide gauges: the use of an improved bathymetry dataset and fine friction parameters adjustment significantly improved our tidal solutions. However, our experiments seem to indicate that the solution error budget is still dominated by bathymetry errors, which is the most common limitation for accurate tidal modelling.

60

## 1. Introduction

The tide impacts on open seas and coastal seas are nowadays largely studied as they influence the oceanic circulation as well as the sediment transport and the ecosystems biogeochemical activity. For instance, Guarnieri et al. (2013) found that tides can influence the circulation by modification of the horizontal advection and can impact on the mixing. According to Gonzalez-Pola et al. (2012) and Wang et al. (2013), tides can also generate strong tidal residual flows by non-linear interactions with the topography. In the South China Sea (SCS), their dissipation can affect the vertical distribution of current and temperature, which in turn might play a role in blooms of the biological communities (Nugroho et al., 2018). The inclusion of tides and tidal forcings in circulation models is therefore not only critical for the representation and study of tides, but also for simulating the circulation and the mixing through different processes: bottom friction modulation by tidal currents, mixing enhanced by vertical tidal currents shear and mixing induced by internal tides, and non-linear interactions between tidal currents and the general circulation (Carter and Merrifield, 2007; Herzfeld, 2009; Guarnieri et al., 2013). Including these mechanisms in circulation models has improved the representation of the seasonal variability of stratifications cycles compared to models without tides (Holt et al., 2017, Maraldi et al, 2013).

At a smaller scale, the effects of tidal currents on the salt and momentum balances in estuaries were first recognized by Pritchard (1954, 1956). Since then, tides are known to play a key role in estuarine dynamics. Affecting mixing, influencing a stronger or weaker stratification depending on the sea water intrusion, and determining the characteristic of the water masses that can interact with the shelf circulation, tides influence is often the main driver of the estuarine dynamics. Amongst others, tidal asymmetry and density gradients are responsible for the presence of estuarine turbidity maximum (mass of highly concentrated suspended sediments, Allen et al., 1980). Slack waters are found to favor sedimentation and deposition, while flood and ebb tend to enhance erosion and resuspension within the estuary, and the tidal asymmetry induce a tidal pumping (i.e. spring tides are more energetic than neap tides). Understanding the dynamics of these turbidity maxima is crucial for harbours and coastal maritime traffic managements as they are often related to high siltation rates,

necessitating regular dredging by local authorities (Owens et al., 2005; Vinh et al., 2018).

These zones of accumulation of suspended sediments are also important for the ecology of coastal areas as sediments can carry pollutants that endanger water quality (Eyre and McConchie, 1993). The ability to understand and predict the formation of these zones related to tide cycles is therefore crucial for coastal and local activities.

The Gulf of Tonkin (from hereafter GoT) covers an area of 115 000 km² from about 16°10'-21°30'N and 105°30-111°E. This crescent-shape semi-enclosed basin, also referred as Vịnh Bắc Bộ in Vietnam or as Beibu Gulf in China, is 270 km wide and 500 km long and lies in between China to the North and East, and Vietnam to the West. It is characterized by shallow waters as deep as 90m and is open to the South China Sea (SCS) through the South of the

Gulf and to the East through the narrow Hainan Strait (Fig. 1a). This latter, also known as Qiongzhou Strait, is on average 30km wide and 50m deep and separates the Hainan Island from the Zhanjiang Peninsula (mainland China). The bottom topography in the GoT and around Hainan Island is rather complex, constantly changing, especially along the coastlines, and partly unknown. Furthermore, the Ha Long Bay area counts about 2000 islets, also

known as notches, sometimes no bigger than a few hundreds square meters.

The GoT is subjected to the South-East Asian sub-tropical monsoon climate (Wyrtki 1961), therefore largely influenced by seasonal water discharges from the Red River (Vietnam) and by many smaller rivers such as the Qinjiang, Nanliu and the Yingzai Rivers (China). The Red River, which brings in average 3500m$^3$/s (Dang et al., 2010) of water along 150km of

115 coastlines, was ranked as the ninth river in the world in terms of sediment discharge in the 1970s with 145-160 Mt/year (Milliman and Meade, 1983). Its sediment supply was drastically reduced since then to around 40 Mt/year of sediments (Le et al., 2007; Vinh et al., 2014). The Red River area accounts for the most populated region of the GoT, with an estimated population of 21.13 millions in 2016, corresponding to an average population

density of 994 inhabitants/km² (from the Statistical Yearbook of Vietnam, 2017). This region is also a key to the economy of Vietnam, with Ha Long Bay (a UNESCO world heritage site) for its particular touristic value, and with the Hai Phong ports system, connecting the North of the country to the world market. This latter is the second biggest harbour of Vietnam with a particular fast-growing rate in terms of volume of cargos passing through the port, of about

4.5x10$^6$ to 36.3x10$^6$ tons from 1995 to 2016, respectively (Statistical Yearbook of Vietnam,

2017). However, the harbour of Hai Phong is currently affected by an increasing siltation due to tidal pumping, related to changes in water regulation by dams since the late 80's. Such phenomenon forces a dredging effort more and more important each year, with 6.6 million US $ spent on dredging activity in 2013 (Lefebvre et al. 2012, Vietnam maritime administration, 2017). In this particular case, fine scale tidal modeling is of great interest for harbour management and risks prevision.

The tides in the SCS and in the GoT have been extensively studied since the 1940s (Nguyen, 1969; Ye and Robinson, 1983; Yu, 1984; Fang, 1986). Skewing through the literature, a lot of discrepancies exist in the cotidal charts before the 1980s, especially over the shelf areas. With the development of numerical models, the discrepancies have been significantly reduced by improving the accuracy of tides and tidal currents prediction.

Wyrtki (1961) was the first to identify the main tidal constituents in the SCS (O1, $K_1$, $M_2$, $S_2$) and Ye and Robinson (1983), the first to successfully simulate the tides in the area. Until recently, only few numerical studies have focused on the GoT (Fang et al., 1999; Manh and Yanaki, 2000; van Maren et al., 2004) and on the Hainan Strait (Chen et al. 2009). By using, for the first time, a high resolution model (ROMS at 1/25°) and a combination of all available data, Minh et al. (2014) gave an overview of the dominant physical processes that characterize the tidal dynamics of the GoT, by exploring its resonance spectrum. This study improved the existing state of the art in numerically reproducing the tides of the GoT, however it also showed the limitations of using a 3D model in representing the tidal spectrum. Indeed, large discrepancies between the model and observations especially for the $M_2$ harmonics and for the phase of $S_2$ were found. The authors explained those discrepancies by the lack of resolution in the coastal areas due to limitations implied by the use of a regular grid and a poorly resolved bathymetric dataset.

The SCS and the GoT are one of the few regions in the world where diurnal tides dominate the semidiurnal tides (Fang, 1986). The tidal form factor (F), or amplitude ratio, defined by the ratio of the amplitude of the two main diurnal over the semi-diurnal constituents (as F=(O1+K1)/(M2+S2)), provides a quantitative measure of the general characteristics of the tidal oscillations at a specific location. If $0< F<0.25$, then the regime is semi-diurnal, if $0.25<F<1.5$, the regime is mixed primarily semi-diurnal, if $1.5<F<3$, the regime is mixed

primarily diurnal and if F>3, the regime is diurnal. Values of F shown on Fig. 2 are calculated using tidal amplitudes from FES2014b-with-assimilation (product described in section 2.2.3). At the entrance of the GoT and at the Hainan strait, the tides are defined as mixed primarily diurnal with F varying from 1.5 and 2.2 depending on the given locations . At the Red River Delta, F is around 15, attesting of a diurnal regime. Indeed, the major branch of energy flux entering the basin from the southwest is weak for the semi-diurnal tides and strong for the diurnal ones. A second branch of energy (also diurnal tidal waves) enters the GoT through the Hainan Strait (Ding et al., 2013).

In coastal seas and bays, tides are primarily driven by the open ocean tide at the mouth of the bay. By resonance of a constructive interference between the incoming tide and a component reflected from the coast, large tide amplitude can be generated. In the GoT case, tidal waves enter the basin from the adjacent SCS and due to the basin geometry, $O_1$ and $K_1$ resonate (Fang et al., 1999). Their amplitudes reach 90 and 80 cm respectively. The Coriolis force deflects to the right the incoming waves and push them against the northern enclosure of the basin. Once the waves are reflected, they propagate southward until they slowly dissipate by friction. Fang (1999) found that the amplitude of the tide gradually decreases from 4 to 2 m North to South during spring tide. The amplitude of $O_1$ in the GoT is larger than $K_1$ because of a larger resonance effect, even though its amplitude in the SCS is smaller than $K_1$ (Minh et al., 2014). The largest semidiurnal waves of the GoT are $M_2$ and $S_2$. They both appear as a degenerated amphidrome with smallest amplitudes near the Red River delta in the northwestern head of the Gulf (between 5 to 15 cm for $M_2$ and below 5 cm for $S_2$) (Hu et al., 2001). Given those values of amplitude, Van Maren et al. (2004) defined the tidal regime in the GoT as mesotidal, and locally even macrotidal, even though diurnal tidal regimes are usually mainly microtidal.

Our first objective in this paper is to propose a robust and simple approach that allows to quantify the sensitivity of the tidal solutions to bottom friction parameterization and to bathymetric changes in the Gulf of Tonkin. This article furthermore represents the first step in a more comprehensive modeling study aiming at representing the transport and the fate of sediments from the Red River to the Gulf using the tridimensional structured coupled SYMPHONIE-MUSTANG model (Marsaleix et al., 2008; Le Hir et al., 2011). In this framework, our final objective is to optimize the configuration (bathymetry) and

190    parameterization (bottom stress) that will be used in this forthcoming study. These objectives are based on the quantification of the response of the tidal solutions to the calibration of the bottom friction and to the improvements of the bathymetry.

        As evidenced by Fontes et al. (2008) and Le Bars et al. (2010), local tidal simulations are
195    mainly affected by the bathymetry and the bottom stress parametrization. These latter often lack of details in remote coastal regions and/or in poorly sampled regions (in terms of bathymetry and tide gauges). It is particularly the case for the GoT. By its location at the boundary between China and Vietnam and by its intense maritime transport activity, the region is extremely difficult to sample, in particular in the highly protected region of Ha Long
200    Bay, in the strait of Hainan and in the nearshore/ coastal areas. In situ data and soundings are consequently rare and yet extremely valuable. The precise goal of the present study is therefore to build an improved bathymetry and coastline database over the GoT and to define the best configuration for bottom stress parameterization in this region, evaluating the impact of those parameters on the tidal representation in the GoT. The resulting optimized
205    configuration will then be used for future numerical studies of ocean dynamics and sediment transport in the region. For that, we first worked on the improvement of the general and global bathymetric datasets available, i.e.: GEBCO (Monahan, 2008), the Smith and Sandwell bathymetry (Smith and Sandwell 1997) and the ETOPO1 Global Relief Model (Amante and Eakins, 2009) by incorporating new sources of data. We then worked on the
210    optimization of the bottom stress parameterization. Our approach to address the issue of the parametrization and to evaluate the impact of our configuration setup is based on the use of the hydrodynamical model T-UGOm model of Lyard et al. (2006). Thanks to its frequency-domain solver, shortly described in the next sections, T-UGOm can indeed perform tidal simulations at an extremely limited computational cost (compared to time-stepping solver), in
215    our case roughly 80 times faster than usual time-stepping hydrodynamical models (i.e. from few minutes for T-UGOm against hours/days). Furthermore, different formulations for the bottom friction can be prescribed as well as a varying spatial distribution of its related parameters (roughness or friction factor). These particular assets allow to perform a large number of sensitivity tests at a reasonable computational cost on bathymetric and bottom
220    stress parametrization, hence to fasten up the processes of precise tuning and calibration/validation of our configuration.

In section 2, we describe the bathymetry, shorelines and waterways construction as well as the numerical model and the modeling strategy in terms of sensitivity experiments. The data used for model evaluation and the metrics used for this evaluation are also presented in this section. In section 3 we present the results regarding the sensitivity of simulations to bottom stress parametrization and to bathymetry. Conclusions and outlook are given in section 4.

## 2. Methods and tools

### 2.1 Shorelines and bathymetry construction

The first step of our work is to improve the shoreline and bathymetry precision. Two global digital shorelines are commonly used for representing the general characteristics of the GoT shorelines: the Global Self-consistent, Hierarchical, High-resolution Geography Database (GSHHG, Wessel and Smith, 1996) and the free downloadable maps from OpenStreetMap (OpenStreetMap contributors, 2015; retrieved from http://www.planet.openstreetmap.org). The GSHHG and OpenStreetMap shoreline products are both superimposed on satellite and aerial images of the GoT downloaded from Bing (https://www.bing.com/maps) and used now as our reference. Bing is chosen here for the accessibility to its opendata, which makes our shoreline construction method do-able by everyone. Fig. 3 shows the shorelines products superimposed on a downloaded image of a small region of the GoT. When closely comparing the shorelines products to the images, it appears that the OpenStreetMap product looks fairly reasonable all along the coastlines of the GoT, except in the Halong Bay area (not shown) where the complex topography and the islets are clearly too numerous. However, the OpenStreetMap shoreline is most of the time shifted by a few meters westwards compared to the land (Fig. 3). The GSHHG dataset suffers from the same problem but shifted by up to 500m eastwards. The observed shifts in both OpenSteetMap and GSHHG products are not documented but could be due, among others, to the use of nautical charts and/or local topography maps for product construction, which could have been collected before accurate GPS measurements in the area. Our objective in this study is to propose a grid matching the reality (i.e. Bing maps, our reference) as close as possible, therefore, none of these databases looked precise enough to meet our expectations.

Consequently, we have built our own shorelines data set, named TONKIN_shorelines, by using the POC Viewer and Processing (POCViP) software (available on the CNRS sharing website, https://mycore.core-cloud.net/index.php/s/ysqfIlcX5njfAYD/download), developed at LEGOS. The satellite and aerial images of the region, previously downloaded from Bing, are georeferenced with POCViP. The software allows the user to draw nodes and segments with a resolution as fine as needed. The resulting TONKIN_shorelines database has a resolution down to 10m and its accuracy is observable on Fig. 3. We followed the same procedure for building a waterways database of the Red River system. This latter is also included in TONKIN_shorelines. For the Ha Long Bay area, another strategy has been considered since drawing by hand each islet would have been unaffordably time consuming. In this case, images from the Shuttle Radar Topography Mission (SRTM) (https://earthexplorer.usgs.gov/) were downloaded and coastlines got extracted and merged to TONKIN_shorelines.

Because of the shallowness of the area, the bathymetry of the GoT is a critical point and could have a strong impact on tidal simulations as it is often the main constraint in tidal propagation (Fontes et al., 2008). The GEBCO 2014 (30 arc-second interval grid) dataset is largely based on a database of ship-track soundings, whose resolution can be locally much finer than 1 km resolution, but gridded data are provided with a ~1 km resolution (as explained on the GEBCO website: https://www.gebco.net/data_and_products/historical_data_sets/. GEBCO dataset can hence be used to represent the slope and the shape of the basin at a relatively large scale (Fig. 1a,b). However, this 1 km resolution is too low to accurately represent detailed geomorphological features, in particular in coastal regions, near the delta and in the Ha Long Bay area. In the purpose of providing an improved tidal solution, we have developed a bathymetry with a better precision, named TONKIN_bathymetry (Fig. 1c,d). For that, we have merged the GEBCO bathymetry with digitalized nautical charts of type CM-93 via OpenCPM (https://opencpn.org/). Note that bathymetric data from nautical charts in coastal shallow areas are often chosen shallower than the "real" bathymetry for navigation security purposes. We also incorporated the tidal flats digital elevation model from Tong (2016). This author used waterlines from Landsat images of 2014 to construct a surface model from elevation contours. As tidal flats are suffering tidal regime with submersion during flood tide and

exposure during ebb tide, their representation is crucial in tidal modelling. TONKIN_bathymetry is merged to TONKIN_shorelines dataset.

This scattered bathymetric dataset shows realistic small-scale structures and depths over the shelf and in the Ha Long Bay area. The details and the islets of the bay are now represented (Fig. 1c,d), as well as the Red River waterways. In the deeper part of the basin, near the boundary, two deeper branches (in light red) are distinguishable. These latter could correspond to the location of the ancient river bed of the Red River during the last glacial time, which split in two around 18°N-108°E (Wetzel et al., 2017). The biggest differences compared to GEBCO are observed in the central part of the region and in the Hainan Strait (Fig. 1e,f). In the strait, the GEBCO bathymetry underestimates depths by roughly 20m (~ 50% in terms of relative difference) compared to TONKIN_bathymetry. In the center, differences can be up to 30m between the two datasets (not shown on the colorbar), corresponding to relative differences up to 100%. Such high observed discrepancies are due to the interpolation of the scattered measuring points from the nautical charts. High relative differences are also observed all along the coastlines, corresponding for most of them to the integration of the intertidal DEM in the Red River delta area, as well as to a better resolution in shallow areas obtained from the nautical charts. Discrepancies in most other parts of the basin remain roughly inferior to 30%. Patches of differences of about 40% between the datasets are also observed at the open ocean boundary of the domain, with GEBCO also underestimating depths in the southernmost part.

We draw attention to the fact that the TONKIN_bathymetry dataset provides an improvement to the available bathymetric dataset, but that some flaws and uncertainties still exist, partly due to sampling methods and shallower waters induced by nautical charts data.

## 2.2  Model, configuration and forcings

### 2.2.1 T-UGOm hydrodynamic model

The tidal simulations are based on the unstructured grid model T-UGOm (Toulouse Unstructured Grid Ocean Model) developed at LEGOS, and is the follow-up of MOG2D (Carrère and Lyard, 2003). In its standard applications, T-UGOm uses unstructured triangle meshes allowing for an optimal grid resolution flexibility, in particular to discretize complex coastal geometry regions, to follow various local dynamical constraints, such as rapid

topography changes or to simply adapt resolution in regions of special interest. The flexibility

of unstructured triangle meshes is fully adequate for fine scale modelling, especially in delta or estuarine systems, whereas usual structured meshes may struggle to represent fine geography of certain areas. The T-UGOm model is widely used in global to coastal modelling, mostly for tidal simulations: in the representation of semi and quarter-diurnal barotropic tides in the Bay of Biscay (Pairaud et al., 2008), in studying the tidal dynamics of

the macro-tidal Amazon estuary (Le Bars et al. 2010), in the representation of tidal currents over the Australian shelves (Cancet et al., 2017) and in assessing the role of the tidal boundary conditions in a 3D model in the Bay of Biscay (Toublanc et al., 2018). Furthermore, T-UGOm has proven its accuracy in global barotropic tidal modelling in the Corsica Channel (Vignudelli et al., 2005) and in a global assessment of different ocean tide

models (Stammer et al., 2014).

In addition to its traditional time-stepping solver, it has the remarkable particularity to include a frequency-domain solver kernel, that solves for the 2D/3D quasi-linearized tidal equations. This spectral mode solves the quasi-linearized Navier-Stokes equations in the spectral domain, in a wave by wave, iterative process (to take into account non-linear effects such as

bottom friction). It has demonstrated its efficiency (accuracy, computational cost) as well for the astronomical tide simulation as for the non-linear tides. The frequency-domain solver can be used either on triangle or quadrangle unstructured mesh, and therefore can be used on any C-grid configuration. Compared to a traditional time-stepping mode that simulates the temporal evolution of the tidal constituents over a given period, the numerical cost of the

frequency-domain mode (2D) is roughly 1000 times smaller.

For our purpose of assessing the sensitivity to various parameters of the tide representation by the model, T-UGO is set up in a 2D barotropic, quadrangle grid, shallow-water and frequency-domain mode (version of the code: 4.1 2616). This configuration (including TONKIN_bathymetry and the specific version of T-UGOm code) is from hereafter

named TKN. The main advantage of this fast and reduced-cost-solver is the possibility to perform in an affordable time a wide range of experiments at the regional or global scale, in order to parameterize the model: optimize bottom stress parametrization, test bathymetry improvements and others numerical developments. In our case, the run duration of a spectral simulation with T-UGOm lasts on average 6 mn (CPU time), which is roughly 40 times

quicker than a simulation with a regional circulation model such as SYMPHONIE (Marsaleix

et al., 2008, CPU time is approximately 4h for a 9-month simulation, corresponding to the required time with SYMPHONIE to separate the tidal waves).

Another useful functionality from T-UGOm for our study is the possibility to locally prescribe the bottom friction, including the roughness length but also the choice of parametrization type. In some shallow coastal regions like the GoT, the presence of fluid mud flow and fine sediments can induce dramatic changes on bottom friction. The quadratic parameterization may be obsolete and a linear parameterization more adequate (Le Bars et al., 2010) and will be tested hereafter. This functionality is essential in those particular regions like shallow estuaries, where the influence of bottom friction on the tides propagation is crucial.

### 2.2.2 Numerical domain over the GoT

The numerical domain over the GoT, built from the TONKIN_bathymetry, is discretized on an unstructured grid made of quadrangle elements (Fig 3). The most commonly used elements in T-UGO are triangles, however here the final goal of our work is to use the resulting grid for coupled hydrodynamical-sediment transport models like SYMPHONIE-MUSTANG using quadrangle structured C-grids. We therefore run the T-UGO tidal solver on a quadrangle grid. As in Madec and Imbard (1996), this grid is semi-analytical. A first guess is provided by the analytical reversible coordinate transformation of Bentsen et al (1999) which produces a bipolar grid. The singularities associated with the two poles are located in the continental mask, slightly to the north of the numerical domain, where the horizontal resolution is the strongest (Fig. 4). This first guess is then slightly modified to control the extension of the grid offshore, in practice to prevent extension beyond the continental shelf. As in Madec and Imbard (1996) this second stage is partly numerical (and preserves the orthogonality of the axes of the grid). The largest edges of the quadrangles are about 5 km at the boundaries of the domain and the smallest of about 150 m long, with a maximum refinement located in the river channels (Fig. 5). This grid allows to represent the complexity of the islets of Ha Long Bay as well as the details of the coastlines of the Red River Delta. A regular C-grid would hardly take into account such complex topography and details.

### 2.2.3  Tidal open-boundary conditions

For modelling barotropic tidal waves, nine tidal constituents have been imposed as open boundary conditions (OBC) in elevation (amplitude and phase) for our domain: $O_1$, $K_1$,

$M_2$, $S_2$, N2, K2, P1, Q1 and M4. Since the astronomical spectrum of tide is dominant in the GoT, 8 out of the 9 constituents simulated are astronomical constituents, and M4 is chosen here as a representative of all non linear interactions. These constituents, ordered by their amplitudes (in the GoT), are the main tidal waves in the GoT and come from the FES2014b global tidal model resolved on unstructured meshes but distributed on a resolution coherent

1/16°x1/16° grid. FES2014b (Carrère et al., 2016) is the most recent available version of the FES (Finite Element Solution) global tide model that follows the FES2012 version (Carrère et al., 2012). The FES2014b global tidal atlas includes 34 tidal constituents and is based on the resolution of the tidal barotropic equations with T-UGOm (frequency-domain solver for the astronomical tides and time-stepping solver for the non-linear tides, described in the

above section). The FES2014b bathymetry has been constructed from the best available (compared to previous FES versions) global and regional DTMs (Dynamical Topography Models), and corrected from available depths soundings (nautical charts, ship soundings, multi-beam data) to get the best possible accuracy, typically 1.3 cm RMS (Root Mean Square error) for the $M_2$ constituent in the deep ocean before data assimilation. The tidal simulation

performed using this configuration and without assimilation is called FES2014b-without-assimilation. Moreover, in addition to the hydrodynamic solutions, altimetry data-derived and tide gauges harmonic constants have been assimilated, using a hybrid ensemble/representer approach, to improve the atlas accuracy for 15 major constituents and fulfill the accuracy requirements in satellite ocean topography correction. This version of FES will from

hereafter be named FES2014b-with-assimilation in the following, in comparison to FES2014b-without-assimilation. Thanks to the accuracy of the prior FES2014b-without-assimilation solutions and the subsequent higher efficiency of data assimilation, this latest FES2014b-with-assimilation version of the FES2014 atlas has reached an unprecedented level of precision and has shown a superior accuracy than any others previous versions (see

http://www.aviso.altimetry.fr/en/data/products/auxiliary-products/global-tide-fes.html and F. Lyard personal comments).

The tidal distribution of the $O_1$, $K_1$, $M_2$ and $S_2$ tidal waves and their first harmonics from FES2014b-with-assimilation and FES2014b-without-assimilation is shown in Figs

4ab,5ab,6ab,7ab, as well as their error along the satellite altimetry track dataset of CTOH-LEGOS (described below in section 2.3.1). FES2014b-with-assimilation shows negligible errors compared to FES2014b-without-assimilation thanks to the assimilation. The main interest of using FES2014b-without-assimilation in our study is to assess the real capacity of the FES model in reproducing the tidal harmonics without using data assimilation, whereas FES2014b-with-assimilation can be used together with satellite altimetry as a reference to evaluate tidal solutions errors.

The T-UGOm code, the model grid and the configuration files used for our simulations are available in Zenodo/Piton (2019a,b,c, see "Code and/or data availability" section).

## 2.3 Simulations and evaluation

We use the model configuration described above to assess the impact of the improvement of our bathymetry database and to optimize the representation of bottom friction in the model. For that we perform sensitivity simulations that we compare with available data using specific metrics. Those tools and methods are presented in this section.

### 2.3.1 Modelling strategy and sensitivity experiments

The T-UGO 2D model (in its frequency-domain, iterative mode) is run on the high-resolution grid described in section 2.2.2. The following sections describe the tests performed for the bottom friction parametrization.

#### 2.3.1.1 Bottom stress parametrization

In shallow areas where current intensities are strong due to a macro-tidal environment combined to strong rivers flows and winds forcing, the sensitivity of the model to the bottom stress is significant. The bottom stress is thus a crucial component for modelling nearshore circulation and sediment transport dynamics (Gabioux et al. 2005; Fontes et al., 2008). The bottom stress formulation depends upon a non-dimensional bottom drag coefficient (or friction coefficient) $C_D$ and can be obtained as follows, in barotropic mode:

$$\tau_b = \rho C_D |\bar{u}| \bar{u} \quad (1)$$

with $\bar{u}$ the depth averaged velocity and $\rho$ the fluid density.


In this study, we test two commonly used parameterizations: a constant drag coefficient $C_D$ assuming a constant speed profile and a drag coefficient $C_D$ depending upon the roughness height $z_0$.

In the first parameterization, a constant profile of the speed is assumed over the whole water height, leading to quadratic bottom stress and a constant $C_D$ that depends on the Chézy coefficient $C$ and on the acceleration due to gravity $g$ (Dronkers 1964):

$$C_D = \frac{g}{C^2} \quad (2)$$


In the second parameterization, a logarithmic profile of the speed is assumed over the whole water column (Soulsby et al. 1993), leading to a $C_D$ depending upon the roughness length $z_0$, the total water height H and the von Karman's constant $\kappa=0.4$:

$$C_D = \left( \frac{\kappa(H-z_0)}{H ln\frac{H}{z_0}+z_0-H} \right)^2 \quad (3)$$

The roughness length $z_0$ (also called roughness height) depends not only on the morphology of the bed (i.e. the presence of wavelets or not) but also on the nature of the bottom sediment. In presence of fluid mud, the friction is considered as purely viscous (Gabioux, 2005). However, the repartition of sediments and the structure of the sea bed are not uniform over the GoT shelf as the Red River discharge causes patches of sediments of different natures (Natural Conditions and Environment of Vietnam Sea and Adjacent Area Atlas, 2007). Consequently, we can expect $z_0$ to vary spatially. This issue can be addressed with T-UGOm since it contains a domain partition algorithm allowing to take into account the spatial variability of the sea bed roughness. Furthermore, this $C_D$ parameterization which includes a logarithmic profile of the speed, allows to adapt the $C_D$ to the model vertical resolution by considering the water column depths, as a way to correspond to the friction coefficient resolution in 3D models.

In the case of fluid mud when the bottom friction is purely viscous and the velocity profiles are linear, Gabioux et al. (2005) described the $\tau_b$ as follows:

$$\tau_b = \rho r \bar{u} \qquad (4)$$

with *r* corresponding here to the friction coefficient.

A third parameterization of the coefficient of friction is tested in this study: a linear profile of the speed is assumed over the whole water column which characterizes viscous conditions. In this case, a linear bottom stress is assumed and *r* depends on the frequency of the forcing

wave $\omega$ (here $O_1$) and the fluid kinematic viscosity *v*:

$$r = \sqrt{\omega} v \quad (5)$$

In this study, these three formulations of the coefficient of friction (Eqs. 2, 3 and 5) are tested

for model parameterization, varying respectively the value of $C_D$, the value of $z_0$ and the value of *r*.

### 2.3.1.2 Sensitivity to uniform friction parameters

Sensitivity numerical experiments were first conducted in order to assess the sensitivity of the

model to uniform parameters of friction for two of the parameterizations described above: a quadratic bottom stress with a uniform drag coefficient $C_D$ (i.e. $C_D$=cst) (Eqs. 1 and 2) and a logarithmic variation of $C_D$ depending on a uniform bottom roughness height $z_0$ (i.e. $C_D$=f($z_0$,H))(Eq. 3). For that, we performed a first set (SET1) of 45 tests running the model with a constant $C_D$ with $C_D$ values spanning from $0.5 \times 10^{-3}$ to $5.0 \times 10^{-3}$ m (see Fig. 9 where we

plotted $C_D$ values spanning from 0.5 to $2.5 \times 10^{-3}$ m). We then performed a second set (SET2) of 6 tests running the model with a $C_D$=f($z_0$,H)) by testing values from $1.0 \times 10^{-1}$ to $1.0 \times 10^{-6}$ m for $z_0$ (see Fig. 10).

### 2.3.1.3 Sensitivity to the regionalization of the roughness coefficient

As mentioned in the previous section, a uniform roughness coefficient does not usually allow for reaching a satisfying level of accuracy over the whole domain, since the variability of the seafloor morphology is not fully taken into consideration. To take this variability into account, the spatial variability of the seabed roughness must be prescribed to the model. For that, our study area is divided into several zones based on seabed sediment

types repartition obtained from the Natural Conditions and Environment of Vietnam Sea and Adjacent Area Atlas (2007).

The third set of sensitivity experiments (SET3, tests A to E in Table 1) consisted in prescribing a linear velocity profile only in the area of fine mud, following Eqs. 4 and 5, with a fixed $r = 1.18 \times 10^{-4}$ m (see Fig 10a), and to test different values of uniform $z_0$ (from $1.0 \times 10^{-2}$

to $1.0 \times 10^{-6}$ m) over the rest of the region, prescribing a logarithmic velocity profile. This value of $r$ is taken from the value empirically tuned on the region of the Amazon estuary and shelf with the configuration described in Le Bars et al. (2010)

The fourth set of sensitivity experiments (SET4, tests 1 to 7 in Table 1) consisted in dividing the region into three zones, according to a supposed spatial distribution of the seabed

sediments, inspired from the above-mentioned Vietnamese atlas (Fig. 12b): zone 1 is mostly composed of muddy sand, zone 2 of mud and zone 3 of sand and coarser aggregates. In each zone, a value of $z_0$ (from $1.0 \times 10^{-2}$ to $1.0 \times 10^{-5}$ m) is prescribed following a $C_D = f(z_0, H)$ (Eq. 3). Note that for this set of experiments every combination of $z_0$ was tested, yet for the sake of clarity we show and describe in the section 4 only the ones with errors (see section 2.3.3)

for $S_2$ solutions below 2.5 cm.

The fifth and last set of experiments (SET5) consisted in dividing the domain into twelve zones, in order to refine the representation of the spatial distribution of the seafloor's sediments following the Vietnamese atlas (Fig. 12c). Zones 1 and 11 correspond to muddy sand; zones 2, 6, 10 and 12 to sand slightly gravel; zones 3 and 5 to sandy mud; zone 4 to fine

mud; zone 7 to sandy gravel; zone 8 to mud slightly gravel; and zone 9 to sand. Different $z_0$ values (varying from $1.0 \times 10^{-2}$ to $1.5 \times 10^{-5}$ m), using $C_D = f(z_0, H)$), were prescribed to each of the 12 twelve zones and corresponding run were performed, each time imposing a random and different value to each zone.

**2.3.2 Satellite data and tide gauges data for model assessment**


The evaluation of the performance of the simulations is made with along-track tidal harmonics obtained from a 19-year (1993 - 2011) long time series of satellite altimetry data available every ten days from TOPEX/Poseidon (T/P), Jason-1, and Jason-2 missions (doi: 10.6096/CTOH_X-TRACK_Tidal_2018_01). These data are provided by the CTOH-LEGOS
(Birol et al., 2016). The tracks of the altimeters passing over the GoT are shown in Fig. 5 to 8, and are spaced by approximately 280 km. To complement those data in the intertrack domain, we also compare our simulations with the FES2014b-with-assimilation tidal atlas, as explained in section 2.2.3.

Harmonic tidal constituents at 11 tide gauge stations are also used for evaluation of the simulations. The data are distributed by the International Hydrographic Organization (https://www.iho.int/)        and        are        available        upon        request        at https://www.admiralty.co.uk/ukho/tidal-harmonics. The name and position of these stations are shown on Fig. 5 a. Amplitudes and phases of O1, K1, M2 and S2 at the 11 gauge stations
are available in Chen et al. (2009).

### 2.3.3 Metrics

For comparison of the simulations with the tidal harmonics from satellite altimetry, two statistical parameters (metrics) are used. These are the root mean square error (RMS*) and
the mean absolute error (MAE). The RMS* computation is based on a vectorial difference which combines both amplitude error and phase error into a single error measure. The errors computations are detailed in Appendix.

### 3. Results

In this section we present the results concerning the sensitivity of the modeled tidal solutions
to the choice of bathymetry dataset and to the choice of bottom friction parameterization. Spatially varying uniform friction parameters only slightly improve the tidal solutions compared to uniform parameters. Furthermore, prescribing a linear parameterization in supposed fluid mud areas does not allow to significantly improve the solutions, unlike in Le Bars (2010). Lastly, the reconstructed bathymetry dataset allows to strongly improve the
semi-diurnal tidal solutions. The improvements consist mainly in a correction near the coasts and in reducing the errors in phase (as can be expected from a bathymetry upgrade). We

present the results of the conducted sensitivity experiments in details in the following sub-sections.

### 3.1 Model sensitivity to bottom stress parameterization

### 3.1.1 Sensitivity to a constant or varying $C_D$ (SET1 and SET2)

We first analyse in this section and in the next one the sensitivity to the parameterization of bottom friction. Firstly, to show the sensitivity to the choice of uniform friction parameters, the model errors (Appendix; Eq. 6) compared to satellite altimetry are shown in Fig. 9

(SET1) and 10 (SET2), for the main tidal constituents ($O_1$, $K_1$, $M_2$, $S_2$) for each values of uniform $C_D$ and $z_0$ tested in SET1 and SET2 described in section 2. On both Figs. 9 and 10, the space in between two solid lines corresponds to the errors for the considered wave (see legend) and the yellow line represents the cumulative errors for the four waves. The dashed red line represents the smallest cumulative error (i.e. the minimum value reached for the

yellow line).

First of all, the diurnal waves $O_1$ and $K_1$ are more affected by the changes in the values of $C_D$ and $z_0$ than the semi-diurnal waves $M_2$ and $S_2$ (Figs. 9 and 10). This can be explained by the fact that diurnal tides are of greater amplitude than semi-diurnal in the Gulf of Tonkin, thus the tidal friction is truly non-linear for $O_1$ and $K_1$ and marginally only for $M_2$ and $S_2$. For $C_D$

values below 0.6 and above $1.0 \times 10^{-3}$ m, $O_1$ and $K_1$ errors are larger than errors for $M_2$ and $S_2$. For example, for $C_D=2.5 \times 10^{-3}$ m the errors for $O_1$ are roughly 4 to 11 times larger than errors for $M_2$ and $S_2$ and errors for $K_1$ are roughly 3 to 10 times larger than errors for $M_2$ and $S_2$, respectively).

Small values of $C_D$ also induce large errors of $O_1$ and $K_1$ (for $C_D=0.5 \times 10^{-3}$ m, errors for $O_1$

are roughly 1.5 to 3.8 times larger than errors for $M_2$ and $S_2$, and errors for $K_1$ are roughly 2.8 to 6.9 times larger than errors for $M_2$ and $S_2$, respectively). High and small values of $z_0$ also trigger larger errors on the diurnal waves (Fig. 10).

Secondly, the tests of sensitivity to a spatially constant friction coefficient $C_D$ show that the lowest error is reached for $C_D=0.9 \times 10^{-3}$ m (the cumulative error is equal to 11.50 cm) (Fig. 9).

This value of $C_D$ is roughly half lower than those used for the whole South China Sea ($2.0 \times 10^{-3}$ m: Fang et al. 1999; Cai et al., 2005) and similar to the one used in the GoT by Nguyen et al. (2014) of $1.0 \times 10^{-3}$ m. The tests of sensitivity to the roughness length $z_0$ show

that the value $z_0 = 1.5 \times 10^{-5}$ m yields the least errors (the cumulative error is equal to 10.96cm) (Fig. 10). This is a relatively small roughness length value, indicating a sea bed composed of very fine particles. Finally, the use of a constant $C_D$ parametrization with a $C_D$ of $0.9 \times 10^{-3}$m or a constant roughness length with a $z_0$ of $1.5 \times 10^{-5}$ m leads to almost identical errors (0.54 cm of differences). The similarity of the results between the two simulations are due to the values of $C_D$ obtained for $z_0 = 1.5 \times 10^{-5}$ m : those values vary spatially from 0.8 to $1.1 \times 10^{-3}$ m, and are thus very close to the optimized value of $C_D = 0.9 \times 10^{-3}$m for a constant $C_D$ (figure not shown). This small spatial variability of the varying $C_D$ explains why the results of the two optimized simulations from SET1 and SET2 are finally similar.

The rather low values of friction ($0.9 \times 10^{-3}$m) and roughness coefficients ($1.5 \times 10^{-5}$ m) suggest the presence of a majority of fine sediments in the GoT. This is consistent with the results from Ma et al. (2010), who found the western and central parts of the GoT to be mainly composed of fine to coarse silts, with a few patches of sand next to Hainan island.

Thirdly, the lowest error for each wave is reached for different values of $C_D$ and $z_0$. In SET1, the lowest error value for $O_1$ is reached when $C_D = 0.9 \times 10^{-3}$ m, while the lowest error value for $K_1$ is reached for $C_D = 1.0 \times 10^{-3}$ m. The lowest errors values of the semi-diurnal waves are reached for $C_D = 1.4 \times 10^{-3}$ m (Fig. 9). In SET2, the lowest errors values of the diurnal waves are reached for $z_0 = 1.5 \times 10^{-5}$ m and are reached for $z_0 = 1 \times 10^{-3}$ m for the semi diurnal waves (Fig. 10). This finding is of course unphysical, and the reader must keep in mind that optimal parameter setting also often deals with model errors numerical compensation. In our study, it is quite obvious that the model bathymetry is far from perfect despite the large efforts carried out to improve the topographic dataset, and remaining errors due to bathymetry imperfections can be partly canceled by the use of an adequate (i.e. numerical, not physical) friction parameter. As bathymetry-induced errors will strongly be affected by the tidal frequency group (species), and since bathymetry directly and distinctly impacts the waves' phase propagation, we can expect that optimal friction parameterization alteration (and corresponding alterations of the bottom shear stress) will slightly vary in a given frequency group but strongly from one to another. The examination of sensitivity studies tends to promote the idea that these differences are mostly due to remaining errors in the bathymetric dataset and the final decision for an optimal friction parameterization will be based on the best compromise for the overall solution errors. As the $K_1$ and $O_1$ sensitivity to friction alteration is prevailing, the compromise is of course mostly driven by these two tidal waves.


To assess the significance of the differences between two parameterizations, Fig. 11 presents spatially the relative differences between the two simulations (SET1 and SET2), in terms of performance in amplitude and phase, taking FES2014b-with-assimilation as a reference. Negative values (in blue) indicate that the simulation from SET2 with a $z_0$ of $1.5 \times 10^{-5}$ m

produces the smallest differences to the reference, while positive values (in red) indicate that the simulation from SET1 with a constant $C_D$ of $0.9 \times 10^{-3}$ m produces the smallest differences to the reference. For K1, values are positive almost all over the GoT basin, indicating that the tidal solution from simulation with a constant $C_D$ (SET1) performs better. However the differences of performance between the two simulations are very small: $\sim 0.5\%$. For O1, M2

and S2 cases, values are mostly negative over the GoT, suggesting that simulation with a $z_0$ of $1.5 \times 10^{-5}$ m (SET2) better represents the tidal solutions for these three waves than simulation with a constant $C_D$ (SET1). Once again however, these improvements are really small (lower than 5%). These results finally show that the tidal solutions are not very sensitive to changes of bottom friction parameterization, from a constant $C_D$ to a $C_D$ varying with $z_0$.

**3.1.2 Sensitivity to the value of spatially varying roughness length (SET3, SET4, SET5)**

The results of the tests performed to assess the model sensitivity to a regionalized roughness coefficient (see Table 1) are shown in Fig. 13.

**3.1.2.1 Sensitivity to a quadratic or linear stress (SET3 vs. SET2)**

No significant improvement of the tidal solutions is obtained from SET3, i.e. by imposing a linear flow in the mud region (where the resolution is the highest), compared to the tests performed with spatially uniform parameters (drag coefficient, SET1 and bottom roughness length, SET2, Figs. 9 and 10). The cumulative error of all four waves (yellow line) is always

above the 10.96 cm value of the smallest error found for SET2 (the smallest cumulative error of 11.33 cm is obtained for test C). Results from SET3 (Tests A to F) show that the solutions still greatly depends upon the roughness length values imposed to the rest of the region, with errors increasing with low and high values of $z_0$ : from tests D to G, cumulative errors increase by a factor 3.5 with $z_0$ values increase from $1 \times 10^{-4}$ to $1 \times 10^{-1}$ m; from test A to B,

errors decrease by a factor 2 with values decreasing from $1 \times 10^{-6}$ to $1 \times 10^{-5}$ m. As previously observed, the diurnal waves $O_1$ and $K_1$ (in tests A to F) are more sensitive to changes in $z_0$

than the semi-diurnal $M_2$ and $S_2$ waves: for $z_0=1x10^{-2}$ m, errors of $O_1$ are 3 and 7 times larger than errors of $M_2$ and $S_2$ respectively, and errors of $K_1$ are 4.5 to 11 times larger than errors of $M_2$ and $S_2$, respectively.

Tests from SET3 suggest that the model sensitivity to bottom friction parameterization in the area of fine mud is limited and therefore poorly influences the cumulative errors over the GoT. This is due to the fact that tidal energy fluxes and bottom dissipation rates are extremely small in this area of fine mud near the Red River delta, as can be seen on Fig. 14 : most of the tidal dissipation occurs along the western coast of Hainan Island and in the

Hainan Strait (values up to -0.2 W m$^{-2}$ in these areas for O1, K1 and M2). Note that the value of $r$ (which is here set to $1.18x10^{-4}$ m following the optimization of Le Bars et al. 2010 over the Amazon shelf) could be tested and could lead to an optimized value for the GoT. However, this would have presumably not significantly affected the final tidal solutions since the choice of a linear parameterization in the area of fine mud did not significantly modify the

tidal solutions.

**3.1.2.2 Sensitivity to a spatially varying roughness length (SET4 and SET5 vs. SET2)**

Improvement of the tidal solutions is obtained from SET4, i.e. by varying spatially the values of the bottom roughness length (imposing a logarithmic speed profile). The cumulative error of all four waves (yellow line) reach a minimum value of 10.43 cm for Test 6 (red dashed

line), which reduced the error found in SET2 by 0.53 cm. This value is reached by imposing values of $z_0=1.5x10^{-5}$ m in regions 1 and 2 (Fig. 12) and $z_0=1x10^{-4}$ m in region 3. Moreover, results from Test 1 and Test 2 show that the solutions largely depend upon the roughness length imposed in region 1 ($z_0$ values of $1x10^{-2}$ to $1x10^{-3}$ m). Again, as already mentioned in

the previous section, the remaining bathymetry-induced errors in our solutions have probably damaged the precise identification of truly physical friction parameterization in this spatially varying roughness length experiment.

The significance of these results is assessed on Fig. 15. Simulation from Test 6 (SET4) better represents K1 harmonics while simulation with varying $C_D$ = f($z_0$ = $1.5x10^{-5}$ m) (SET2) better

represents O1 harmonics, taking FES2014b-with-assimilation as a reference. However the relative improvements from a simulation to another are again very small (<3%). Considering the semi-diurnal waves, the differences are heterogeneous over the basin, with overall a slightly better representation (difference of ~2%) of the waves by the simulation with $C_D$ =

f($z_0 = 1.5\times10^{-5}$ m) (SET2). These results finally suggest that differences between simulations from SET2 and SET3 are locally and globally insignificant, and that the tidal solutions in the GoT are therefre not very sensitive to changes of bottom friction parameterization, from a constant $z_0$ to a spatially varying $z_0$.

Lastly, the results of the tests from SET5 did not show any improvement on the tidal solutions, compared to SET3 and SET4. The minimum cumulative error found in SET5 is reached by imposing $z_0$ values that correspond exactly to the configuration of Test 6 in SET4 and is consequently also equal to 10.43 cm (i.e. the same as in SET4). This result again suggests that the model seems to be insensitive to high spatial refinement of the bottom sediment composition and associated roughness for the representation of tidal solutions.

### 3.2 Sensitivity to the bathymetry

Model bathymetry is a key parameter for tidal simulations. In order to evaluate the sensitivity of the model to the bathymetry, an additional sensitivity simulation is performed. First, the solutions obtained with the grid configuration with improved bathymetry and shoreline datasets described in section 2.1 (Fig. 4) and a spatially varying roughness length (described in Test 6 from SET2, Table 1) with a logarithmic velocity profile are chosen as this choice of bottom roughness parameterization has shown the best tidal solutions (the least errors to satellite altimetry) in section 3.1.2. This simulation is named TKN hereafter. Second, we run a twin simulation with exactly the same configuration, parameterizations and choice of parameters, except that the bathymetry and shoreline are not built from our improved dataset but from the default GEBCO bathymetry dataset and default-shoreline dataset. This simulation is hereafter named TKN-gebco. The results from these tests are presented in this section, where we evaluate the quality of the tidal solution obtained in our different simulations.

### 3.2.1 Average assessment over the domain

We first evaluate the tidal solution in average over the domain. Integrated alongtrack RMS* errors (Appendix; Eq. 6) between modelled and altimetry derived ocean tide harmonic constants (noted hereafter AH for Altimetric Harmonic) are shown in Fig. 16 a. FES2014b-

with-assimilation errors are globally always lower than errors given by the three other
       simulations, thanks to the assimilation of the satellite altimetry: 3 to 4.7 times lower for $O_1$,
       2.6 to 3.2 times for $K_1$, 2.3 to 3.7 times smaller $M_2$ and 1.3 to 2.7 times for $S_2$. As explained
       in section 2.2.3, FES2014b-with-assimilation, which minimizes the error, is used in the
       following as a reference for the evaluation of our simulations to spatially complement
altimetry data which are only available along the altimetry tracks.

       Alongtrack RMS* errors for $M_2$ are reduced by 12% in TKN-gebco relative to FES2014b-
       without-assimilation and by 47% in TKN. The errors are also lower for $S_2$, with a reduction
       of the errors by 41% between FES2014b-without-assimilation and TKN-gebco, and by 56%
       between FES2014b-without-assimilation and TKN. On the other hand, both TKN and TKN-
gebco simulations show bigger errors than FES2014b-without-assimilation for $O_1$ and $K_1$.
       However, TKN simulation increases the errors for $O_1$ by 7% relative to FES2014b-without-
       assimilation whereas TKN-gebco increases the errors by 58%. The complex errors for $K_1$
       obtained from both TKN and TKN-gebco simulations increase by roughly 12 and 20%
       compared to FES2014b-without-assimilation, respectively. Such results illustrate the fact that
$K_1$ wavelengths are longer than the other waves' wavelengths considered here, e.g.: at 60 m
       depth, $K_1$ wavelength is 2000 km and $M_2$ wavelength is 1000 km (Kowalik and Luick, 2013),
       therefore $K_1$ is less sensitive to bathymetric variations. These results further illustrate FES
       model efficiency in tidal simulation in coastal areas, which is related in particular to the use
       of an unstructured triangle grid mesh specifically dedicated to finely adapt to complex coastal
topography and coastline.

       The mean absolute differences MAE (Appendix; Eq. 7) of amplitude and phase of the four
       tidal constituents between our simulation TKN and AH, and between the two FES2014b
       products and AH are given in Table 2. The errors to AH in both amplitude and phase are
always reduced in TKN compared to FES2014b-without-assimilation (except for the phase of
       S2): from 1.2 times for the phase of K1 to 2.7 times for the amplitude of S2. However, errors
       to AH are increased in both amplitude and phase in TKN compared to FES2014b-with-
       assimilation (except for the amplitude of S2).

In addition, integrated RMS* errors between simulated and observed tidal harmonics from
       tide gauges are presented in Fig. 16 b. Again, FES2014b-with-assimilation errors are globally

always lower (for O1, K1 and M2) than errors given by the three other simulations, thanks to the assimilation of including those tide gauges data.

TKN-gebco increases the errors for O1 by 27% relative to FES2014b-with-assimilation while TKN increases it by 16%. TKN also slightly reduces (by 0.2 cm)  the integrated tidal gauge RMS* errors for O1 compared to FES2014b-without-assimilation. The complex errors for K1 obtained from both TKN and TKN-gebco simulations increase by roughly 25% and 90% compared to FES2014b-with-assimilation, respectively. The errors for M2 are increased by 11% with TKN and by 101% with TKN-gebco compared to FES2014b-with-assimilation. Similar to O1 case, TKN slightly reduces (by 0.3 cm) the  RMS* errors for M2 compared to FES2014b-without-assimilation. Finally, both TKN and TKN-gebco simulations reduces the errors for S2 by 44% and 9% compared to FES2014b-with-assimilation, respectively. So for the 4 main tidal constituents, errors between simulated and observed tidal harmonics from tide gauges are significantly reduced in TKN compared to TKN-gebco.

These results show first that TKN configuration brings a clear improvement in tidal solutions compared to TKN-gebco configuration, and second that it only slightly improves tidal solutions for some of the tidal components compared to FES2014b-without-assimilation. This last result related to the use of an unstructured grid in FES2014b (with and without-assimilation), better adapted for the representation of the complex coastal topography that the structured grid that is used to optimize our TKN configuration since it will be used in our tridimensional structured grid model.

**3.2.2 Spatial assessment of tidal solutions**

The modelled $O_1$, $K_1$, $M_2$ and $S_2$ fields for TKN, TKN-gebco, FES2014b-without-assimilation and FES2014b-with-assimilation are shown in Figs. 5 to 8. For each tidal component and simulation, the complex errors RMS* between these simulations results and AH are represented for each point of the altimetry track by the circles superimposed on the maps.

Both model simulations (TKN-gebco and TKN) reproduce well the distribution patterns of $O_1$ harmonics compared to FES2014b-with-assimilation, improving the results compared to FES2014b-without-assimilation. Moreover, TKN solutions look more accurate than TKN_gebco. Errors to AH for $O_1$ (circles on the maps) are smaller than 10 cm in TKN_gebco

and are reduced by 35% compared to FES2014b-without-assimilation. They are further reduced in TKN: most of the errors are smaller than 5 cm and are reduced by 50% compared to FES2014b-without-assimilation (Fig. 5). Note that higher errors of AH (of about 20cm) are observed in TKN and TKN-gebco in the Hainan Strait and also near the coasts. These errors also appear in FES2014b-without-assimilation and, to a lesser extent (with values of about 15 cm in the Hainan Strait) in FES2014b-with-assimilation. The increase of complex errors in these particular areas could be explained by either model errors associated with errors in coastal bathymetry and shorelines, whose accuracy is decisive to shallow water tidal wave and/or to erroneous altimetric data (land contamination in the altimeter footprint). For $K_1$, even though both model simulations reproduce well the distribution pattern of the harmonics compared to FES2014b-with-assimilation, the errors to AH compared to FES2014b-without-assimilation are not reduced in TKN-gebco nor in TKN (Fig. 6). Errors to AH for $K_1$ are equals or smaller than 10 cm in FES2014b-without-assimilation, TKN-gebco and TKN along the altimetry tracks, and are extremely similar between those simulations. As observed for $O_1$, larger and similar errors of about 20 cm are also observed in the Hainan Strait in TKN-gebco and TKN and in the two FES2014b products (though with smaller values of about 15 cm in FES2014b-with-assimilation that includes assimilations). Furthermore, the angle of the circles' radius indicates which of the error in amplitude or the error in phase dominates the complex error. The smaller the angle is to the ordinate axis, the more the error in phase dominates the complex error. On the contrary, the bigger the angle is to the ordinate axis, the more the error in amplitude dominates the complex error. When the angle to the ordinate axis approaches 45°, the error in phase and the error in amplitude account equally for the complex error. For both $O_1$ and $K_1$, errors in phase are dominating the northern and central parts of the region, while the phase and amplitude account equally for the complex errors in the southernmost part of the region and in the Hainan Strait (Fig 4cd;5cd).

Fig. 7 shows that $M_2$ amplitude is globally overestimated by TKN and TKN-gebco compared to FES2014b-with-assimilation and FES2014b-without-assimilation, especially in the areas where the wave resonates (i.e. in the north eastern bay and in the south-western part). Differences to FES2014b-without-assimilation are up to 30 to 35 cm in TKN and in TKN-gebco (in the north-eastern bay), with amplitudes increasing by almost 45% in model simulations. Both model simulations increase the resonance of $M_2$ in the bays. These

amplitude overestimations could be partially explained by the bathymetry dataset (TONKIN_bathymetry) which integrates nautical charts that underestimate depths, especially in shallow areas, for navigation purposes.

However, the amplitudes are underestimated near the amphidromic point (20.7°N, 107.3°E) by both simulations compared to FES2014b-without-assimilation (by up to 10 cm, roughly 80%). Globally, errors to altimetric harmonics are reduced in TKN by 30% compared to TKN-gebco: most of the errors in TKN are smaller than 10 cm while most of the errors in TKN-gebco are equals or larger than 10 cm. Both simulations show also smaller errors to AH south of Hainan Island compared to FES2014b-without-assimilation by up to 50%, with errors of about 1 cm in FES2014b-without-assimilation and of about 0.5 cm in model simulations). Again, large errors are observed in the Hainan Strait in simulations and both FES2014b products, however the errors are reduced by 25% in TKN and TKN-gebco compared to FES2014b-without-assimilation. Furthermore, these errors are dominated by errors in phase rather than in amplitude.

Lastly, both TKN and TKN-gebco model simulations overestimate the amplitude of $S_2$ compared to FES2014b-with-assimilation and FES2014b-without-assimilation by up to 10 cm (approximately 50%) in the south-western and north-eastern bays, where $S_2$ resonates (Fig. 8). The resonance of $S_2$ in the bays is therefore amplified in both model simulations. These amplitude amplifications observed in both models could be again partly explained by the bathymetric soundings of TONKIN_bathymetry that underestimate depths in shallow water areas. Similar to $M_2$ case, both simulations also underestimate by roughly 5 cm the amplitude of $S_2$ near the amphidromic point (20.7°N, 107.3°E) compared to the two FES2014b products. However, the complex errors to AH are globally 50% smaller in TKN (between 2 to 3 cm) than in TKN-gebco (between 4 to 5 cm). Furthermore, errors to AH remain large in the Hainan Strait (up to 6 cm) in both FES products and both model simulations. In the very near coastal areas, errors to AH in TKN and TKN-gebco are larger than in the rest of the basin (up to 6 cm). However, the errors to AH in TKN are reduced by 30% in the south-western most part of the region. Like $M_2$ case, the complex errors of $S_2$ are dominated all over the basin the by errors in phase rather than errors in amplitudes.

### 3.3 Assessment of tidal solutions with previous studies

The mean absolute differences MAE (Appendix; Eq. 7) of amplitude and phase of the four tidal constituents between our simulation TKN and the satellite altimetry are given in Table 2. Our results are compared with the errors given in Minh et al. (2014) and in Chen et al. (2009). Minh et al. (2014) authors used a ROMS_AGRIF simulation at a resolution of $1/25°$ x $1/25°$ over the GoT, and compared their solutions to the same altimetric dataset as the one used in our study. Chen et al. (2009) compared their simulations, performed with ECOM (Extended Control Model) at a 1.8x1.8 km resolution (covering the area 16°N-23°N, 105.7°E-114°E), to gauge stations located along the GoT coast. Our simulation TKN shows large improvements in both amplitude and phase for the four constituents (except for the phase of $M_2$) compared to the results of Minh et al. (2014). The errors are reduced by approximately 60% for the amplitudes of $M_2$ and $S_2$ approximately, and by roughly 40% for the amplitudes of $K_1$ and $O_1$, respectively. The errors in phase for $O_1$ and $K_1$ are also reduced by approximately 50% in our simulation. Our results show also improvements compared with the errors proposed by Chen et al. (2009), for both amplitude and phase of $S_2$ (by 65% in amplitude and by 35% in phase), $O_1$ (by 50% in amplitude and almost 60% in phase) and $K_1$ (by 68% in amplitude and by 41% in phase). Only the solutions of $M_2$ are not improved by our simulation compared to Chen et al. (2009): they remain the same in amplitude and increase by 14% in phase.

The improvement of our tidal solution from TKN compared to these two previous studies could be due first to the use of T-UGO that is specifically developed for tidal modelling purpose, compared to models used by Minh et al. (2014) and Chen et al. (2009), which are hydrodynamical model not specifically conceived for tidal representation. Second, to our model configuration which has been specifically optimized for tidal modelling purpose, in terms of grid resolution, bathymetry accuracy and resolution and bottom friction parameterization.

**4. Conclusions**

This study takes place in the framework of a more comprehensive modeling project which aims at representing the transport and fate of the sediments from the Red River to the GoT. In this future study, the ocean dynamics and the sediments transport will be represented using the regional circulation model SYMPHONIE coupled with the sediment model MUSTANG.

As tides have a major effect on the sediment dynamics within the estuaries and in the plume area (Pritchard 1954, 1956; Allen et al., 1980; Fontes et al., 2008; Vinh et al., 2018), it is necessary to accurately represent the tidal processes before investigating the fine scale sediment physics. This coupled-model will allow for example to study the impact on tides of freshwater discharges and its strong seasonal variability, since this effect could be relatively

important in the very nearshore coastal area.

The optimization of the configuration and parameterizations of this coupled tridimensional structured grid model is the final objective of this study. Optimizing the bathymetry and the parameterization of the bottom shear stress is crucial in shallow-water regional and coastal modeling since they both are critical parameters influencing the propagation and distortion of

the tides (Fontes et al., 2008; Le Bars et al., 2010). The T-UGO hydrodynamical model is used in this study in its spectral mode, which allows the user to perform fast and low-cost tests (compared to simulations with sequential models like SYMPHONIE) on various configurations. This strategy allows to assess and quantify the importance of each element considered and to determine the best configuration that will be applied in the above-

mentioned forthcoming modeling study with SYMPHONIE-MUSTANG model.

In this study, we have first constructed an improved bathymetric dataset for the region of the GoT from digitalized nautical charts, soundings, intertidal DEM and GEBCO bathymetry dataset. We also integrated to this bathymetry a new coastline dataset created with POCViP

and satellite images, since the existing descriptions of the GoT coastlines, the Ha Long Bay islets and the Red River delta were very poor. We then performed tests with the fast-solver 2D T-UGOm model on an unstructured grid refined in the Red River delta and Ha Long Bay area to test the added value of this improved bathymetric dataset. With this new bathymetry, we have been able to reduce the errors (taking alongtrack altimetry data and tide gauges data

as a reference) of the representation of $M_2$ and $S_2$ in T-UGOm simulations by 40% and 25% respectively and $O_1$ and $K_1$ by 32% and 6% respectively, compared to simulations that use the regular GEBCO dataset. Our improved bathymetry showed also better solutions for the semi-diurnal waves than the tidal atlas FES2014b_hydrodynamics (errors reduced by 47% or $M_2$ and by 25% for $S_2$), even though our model seems to amplify their resonance. Moreover,

our simulations also improved accuracy over the existing state of the art, by reducing the errors in amplitude of the semi-diurnal waves by 60%, the errors in amplitude of the diurnal

waves by 40%, and the errors in phase of the diurnal waves by 50% compared to the results found by Minh et al. (2014). We believe the remaining errors in our best tidal solutions are due to potential lacks of details and resolution in the bathymetry. Since bathymetry directly

impacts the waves' speeds, bathymetric uncertainties may lead to alterations of the bottom shear stress.

The other key parameter influencing shallow-water tidal modelling is the bottom friction. In this study, the use of a constant $C_D$ parametrization or the use of a $C_D$ depending on the

roughness length led to fairly similar results, in line with the results found by Le Bars et al. (2010) in the Amazon estuary. Furthermore, our study shows that the model is very sensitive to the values imposed to $C_D$ and $z_0$, especially for the diurnal waves with errors increasing for extreme $C_D$ and $z_0$ values. The lowest cumulative errors of all four waves (of 11.50 and 10.96cm) were found for a uniform $C_D$ of $0.9 \times 10^{-3}$ m (prescribing a constant velocity profile)

and for a uniform $z_0$ of $1.5 \times 10^{-5}$ m (prescribing a logarithmic velocity profile), respectively. More importantly, the regionalisation of the roughness length into three regions, for addressing the issue of representing the complexity of seabed composition and morphology, only slightly improved the accuracy of our simulation, with a lowest cumulative error for all four waves of 10.43 cm. Finer local adjustments of the roughness length or the choice of a

linear velocity profile in the area of fine mud, did not significantly improve the accuracy of our simulations. In particular, the model in this configuration showed a very limited sensitivity to the presence of fine mud and a greater sensitivity to the roughness length values prescribed in the rest of the region, which was unexpected following the results of Le Bars et al., 2010 and explained by the low energy dissipation occurring in this area of fine mud.


Our results therefore quantitatively showed that the key parameter for the representation of tidal solutions over a shallow area like the GoT is the choice of bathymetry and shoreline dataset. Second, they revealed that the choice of the bottom stress parameterization does not significantly affect the performance of the model, but that, for a given parameterization, the

choice of the value of the friction coefficient is important. Furthermore, the use of T-UGOm in a 2D barotropic mode showed its efficiency in tidal spectral modelling with reduced simulation durations in both CPU and running times compared to structured grid numerical models. This allowed us to optimize our configuration in terms of grid, bathymetry and

bottom friction parameterization regarding the representation of the tidal solutions. Our resulting configuration brought a clear improvement in the tidal solutions compared to previous 3D simulations from the literature. The modeling strategy proposed here thus showed its efficiency in quickly optimizing the configuration that will be used in future works to address the issue of sediment transport and fate in the GoT, which was our primary objective. Finally, our configuration did not produce a clear improvement compared to FES2014b-without-assimilation, and perform worse than FES2014b-with-assimilation, due to the use in both FES2014b simulations of an unstructured grid better representing the coastal topography complexity, and the use of assimilation in FES2014b-with-assimilation. This underlines the importance of data assimilation for the production of tidal atlases, and the need to go on developing satellite mission and in-situ campaigns, despite the great improvements of numerical models in the last decades.

The evaluation of T-UGO performances should be completed with the study of tidal currents, which is for now limited in the area due to in-situ data unavailability, but will be possible in the future thanks to the on-going development of observed current datasets from high-frequency radars (Rogowski et al., 2019). Note also that the use of nautical charts for bathymetry construction could have led to overestimation of the tidal amplitudes in coastal areas, due to underestimation of real depths for navigation safety purposes. Furthermore, using bathymetry data available from digitalized navigation charts was a relatively simple way (compared to performing additional in-situ measurements) to significantly improve the representation of topography in the coastal and estuarine areas of the GoT, and could be applied successfully in other regions. However, updates and improvements in shorelines and bathymetry databases, particularly in the river channels, coastal areas and in the Hainan strait would still improve the present results and especially reduce the tendency to increasing errors at the coasts. Continuous efforts should be made in bathymetric data acquisition and sharing them with the community should be a crucial concern.

**Code and/or data availability:**

The model grid, which integrates the bathymetry and the coastline datasets developed in this study, is available in Zenodo/Piton (2019a). The T-UGO model code installation instructions are updated at

ftp://ftp.legos.obs-mip.fr/pub/ecola/README.html, and the code, as well as the updated tools and the

poc-solvers are available on   https://hg.legos.obs-mip.fr/tools/. An archive of the exact version of T-

UGOm used in this study (version 2616:78a276dd7882) is also available in Zenodo/Piton (2019b).

The  configuration  files  (initial  conditions  and  modified  drag  coefficients)  are  available  in

Zenodo/Piton (2019c). Model boundary conditions (i.e. FES2014b products) are available through:

https://www.aviso.altimetry.fr/en/data/products/auxiliary-products/global-tide-fes.html.  The  satellite

altimetry  track  dataset  of  CTOH-LEGOS  for  model  outputs  comparison  are  available

through:http://ctoh.legos.obs-mip.fr/products/coastal-products/coastal-products-1/sla-1hz          (doi:

10.6096/CTOH_X-TRACK_Tidal_2018_01).

**Authors contribution:**

VP, MH, SO and FL designed the experiments and VP carried them out. FL developed the model

code and VP performed the simulations. VP, TH and DA constructed the bathymetry dataset. VP

prepared the manuscript with contributions of MH, FL, SO and PM.

**Acknowledgments:**

Map      data      copyrighted      OpenStreetMap      contributors      and      available      from

https://www.openstreetmap.org.  We  thank  the  CTOH  team  of  LEGOS  (Toulouse)  for

providing coastal altimetry data. This paper is a contribution to the LOTUS International

Joint Laboratory (lotus.usth.edu.vn).

**Appendix:**

For comparison of the simulations with the tidal harmonics from satellite altimetry, we first

introduce the vectorial difference z, or complex difference, as:


$$z = z_m - z_o \qquad (5)$$

with $z_m = A_m e^{iGm}$ the vector representing a given modelled tidal constituents (of amplitude $A_m$

and phase $G_m$) and $z_o$ the vector representing the observed tidal constituent.


For assessing the errors between the simulations (modelled constituents) and the altimetry (observed constituents), we compute the root mean square error (RMS*), like in Stammer et al. (2014) and in Minh et al. (2014). RMS* depends upon the vectorial difference z and is computed for each given constituent of each simulation, as follows:


$$RMS = \sqrt{\left(\frac{1}{N}\sum_{i=1}^{N} 0.5\,|z|^2\right)}$$

$$RMS = \sqrt{\left(\frac{1}{N}\sum_{i=1}^{N} 0.5\left[\left(A_m cos(G_m) - A_o cos(G_o)\right)^2 + \left(A_m sin(G_m) - A_o sin(G_o)\right)^2\right]\right)} \quad (6)$$

with $A_m$ and $G_m$ being respectively the amplitude and phase of the modelled constituent, $A_o$ and $G_o$, the amplitude and phase of the constituent from satellite altimetry and N the number

of points of comparison (i.e. the number of equivalent gauge stations along the altimetry tracks).

The model performance is also estimated using the mean absolute error (MAE). MAE measures the mean of the difference between the simulated and the observed values and is

computed for each constituent according to:

$$MAE = \frac{\sum_{i=1}^{N}|E_i|}{N} \quad (7)$$

with $E_i$ representing for each point i of the track the difference between the modeled constituent and the observed constituent. The MAE is separately calculated for amplitudes

and for phases.

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

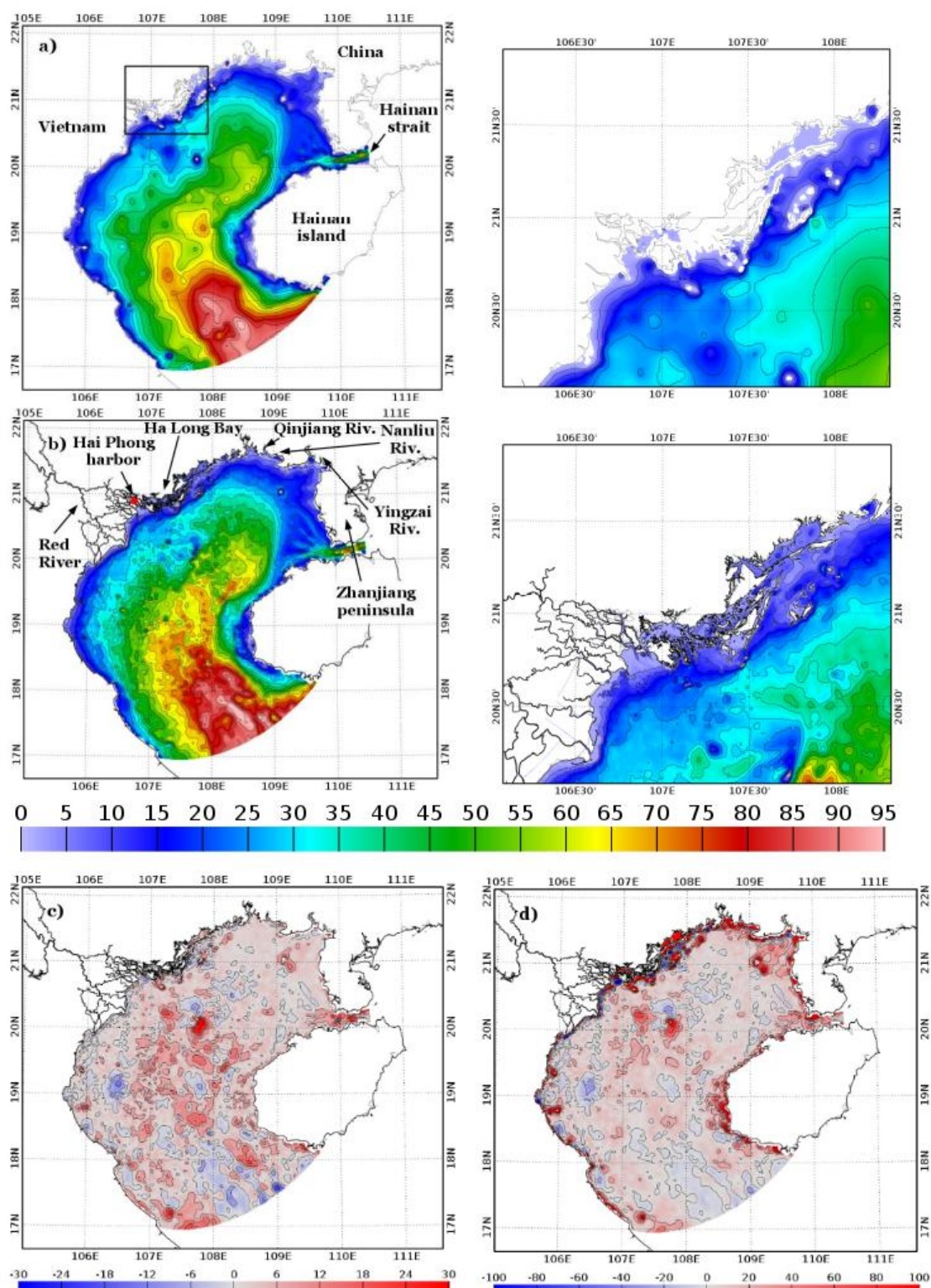

**Figure 1: (a, left) Gebco bathymetry (in m) and (a, right) details of the Ha Long Bay area (black rectangle in a,left). (b, left) TONKIN_bathymetry data set merged with TONKIN_shorelines over GoT and (b,right) zoom in the Ha Long Bay area. (c) Absolute (m) and (d) relative (%) differences between TONKIN_bathymetry and Gebco bathymetry (in m).**

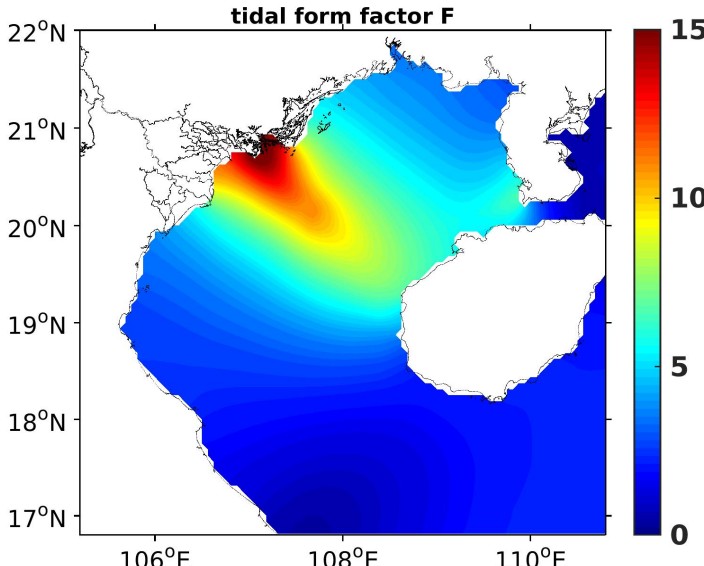

**Figure 2: Map of tidal form factor F computed with the amplitudes of tidal waves O1, K1, M2 and S2 obtained from FES2014b-with-assimilation. 0<F<0.25 corresponds to semi-diurnal regime, 0.25<F<1.5 mixed primarily semi-diurnal, 1.5<F<3 mixed primarily diurnal and F>3 diurnal regime.**

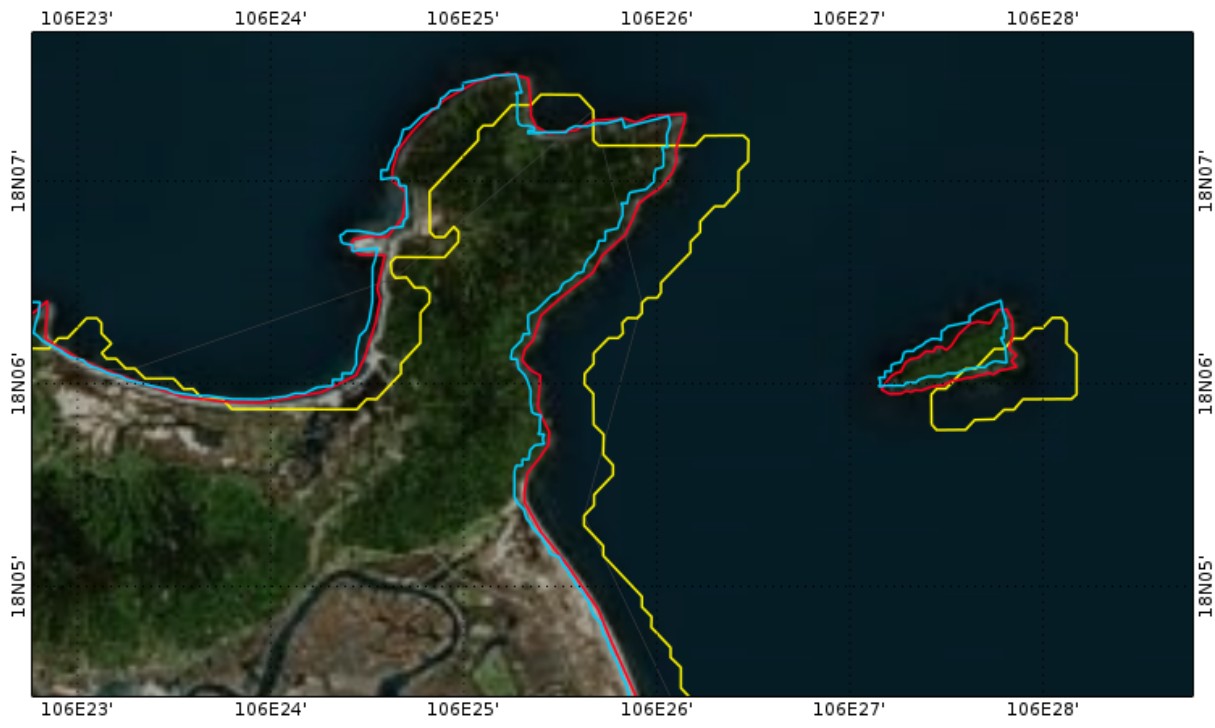


**Figure 3: Shorelines products from OpenStreetSap (blue line), GSHHS (yellow line) and TONKIN_shorelines (red line) superimposed on a satellite image downloaded from**

**Bing (Bing<sup>TM</sup> Copyright) over a small region of the GoT.**



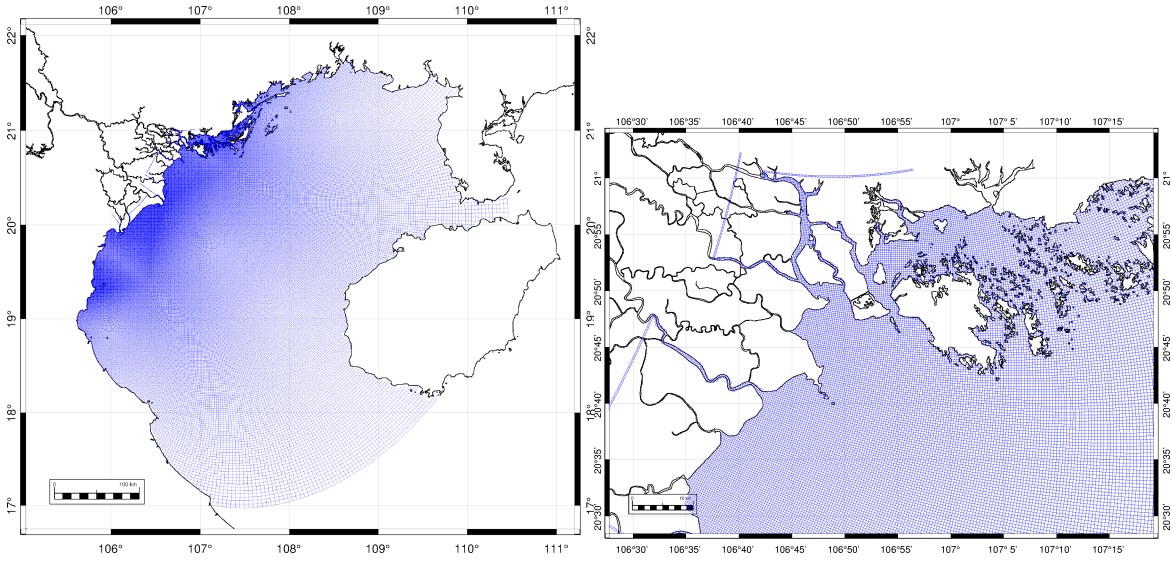


**Figure 4:  Model mesh over the GoT (left) with a zoom in Halong Bay region (right). The maximum refinement (150 m) is reached in the river channels.**


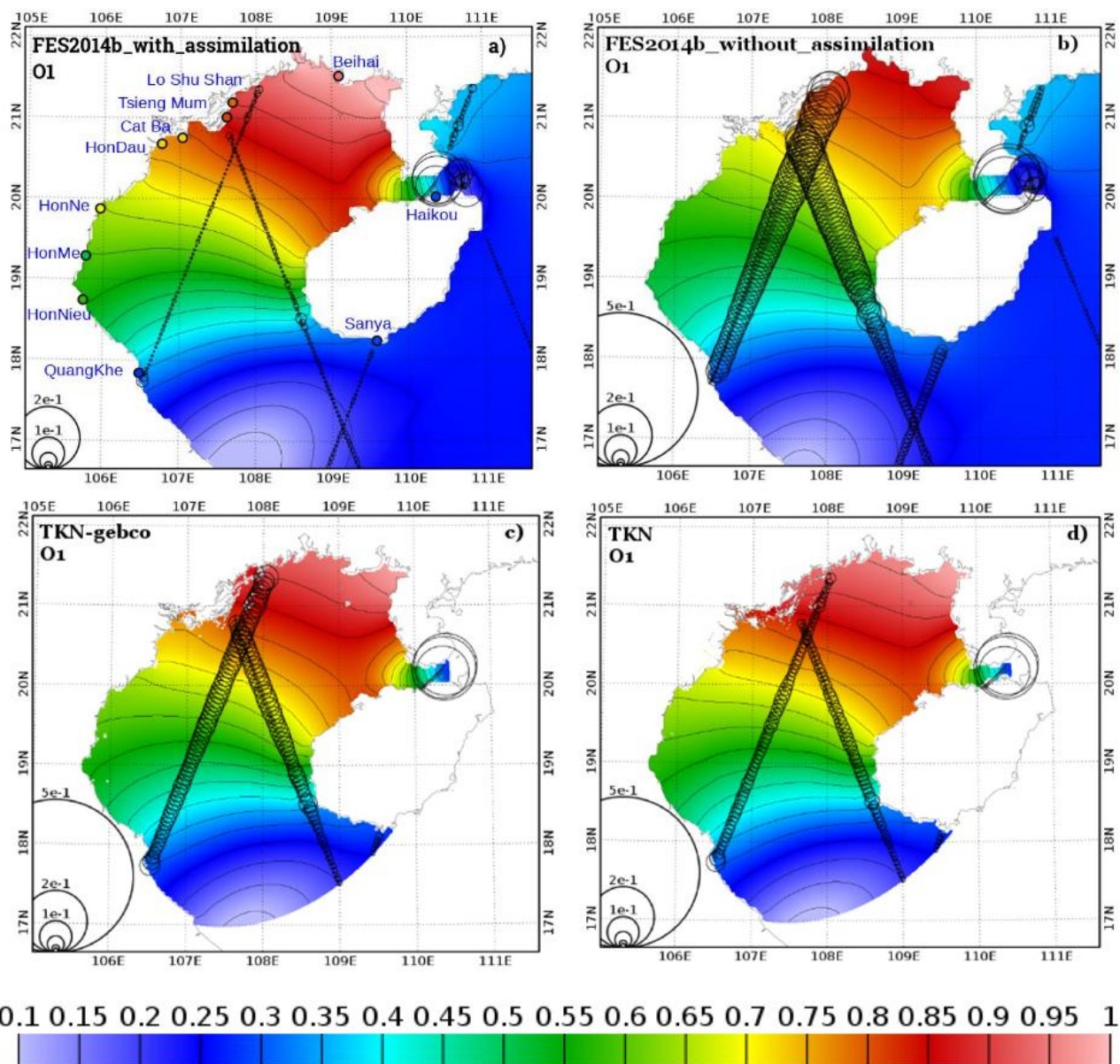

**Figure 5: O1 tidal amplitude (in m) from different products: (a) FES2014b-with-assimilation, (b) FES2014b-without-assimilation, (c) TKN-gebco and (d) TKN. The circle diameter is proportional to the complex error (Appendix ; Eq. 6 ) between the solutions and satellite altimetry (in m). The colored circles denote the amplitude of O1 harmonic measured at the corresponding tide gauge station.**

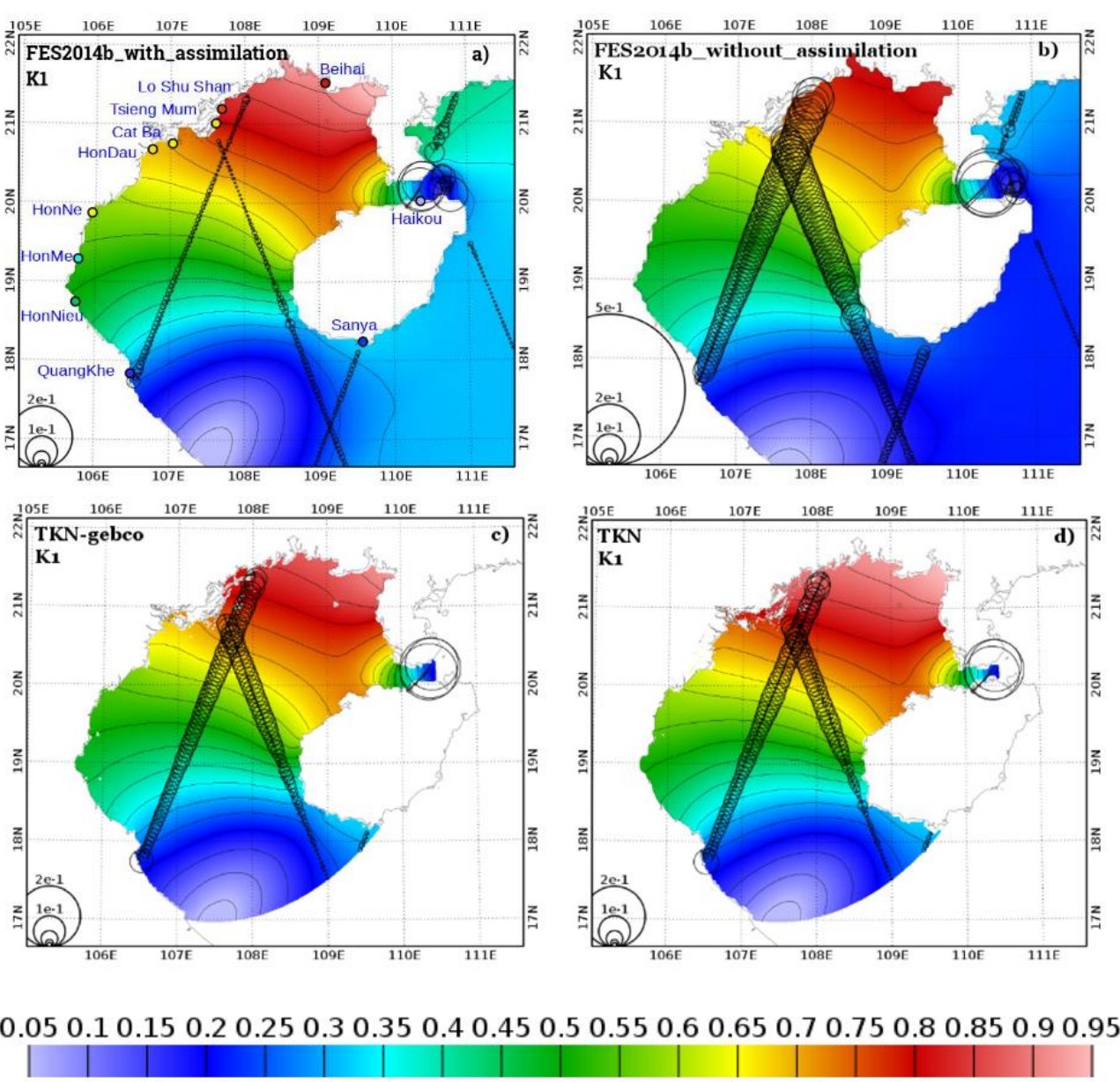


**Figure 6: Same as Fig. 5 for K1.**


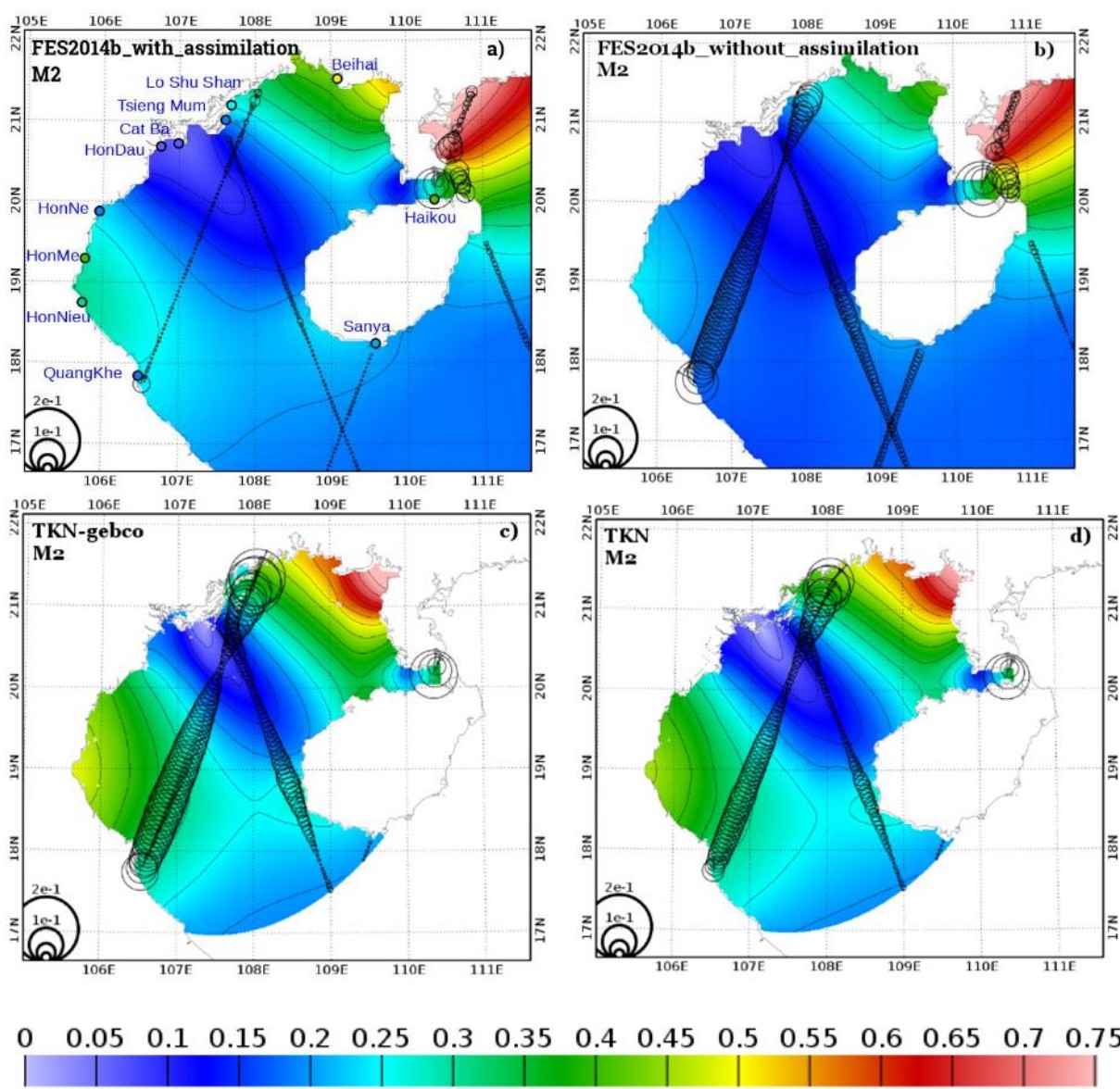

**Figure 7: Same as Fig. 5 for M2.**


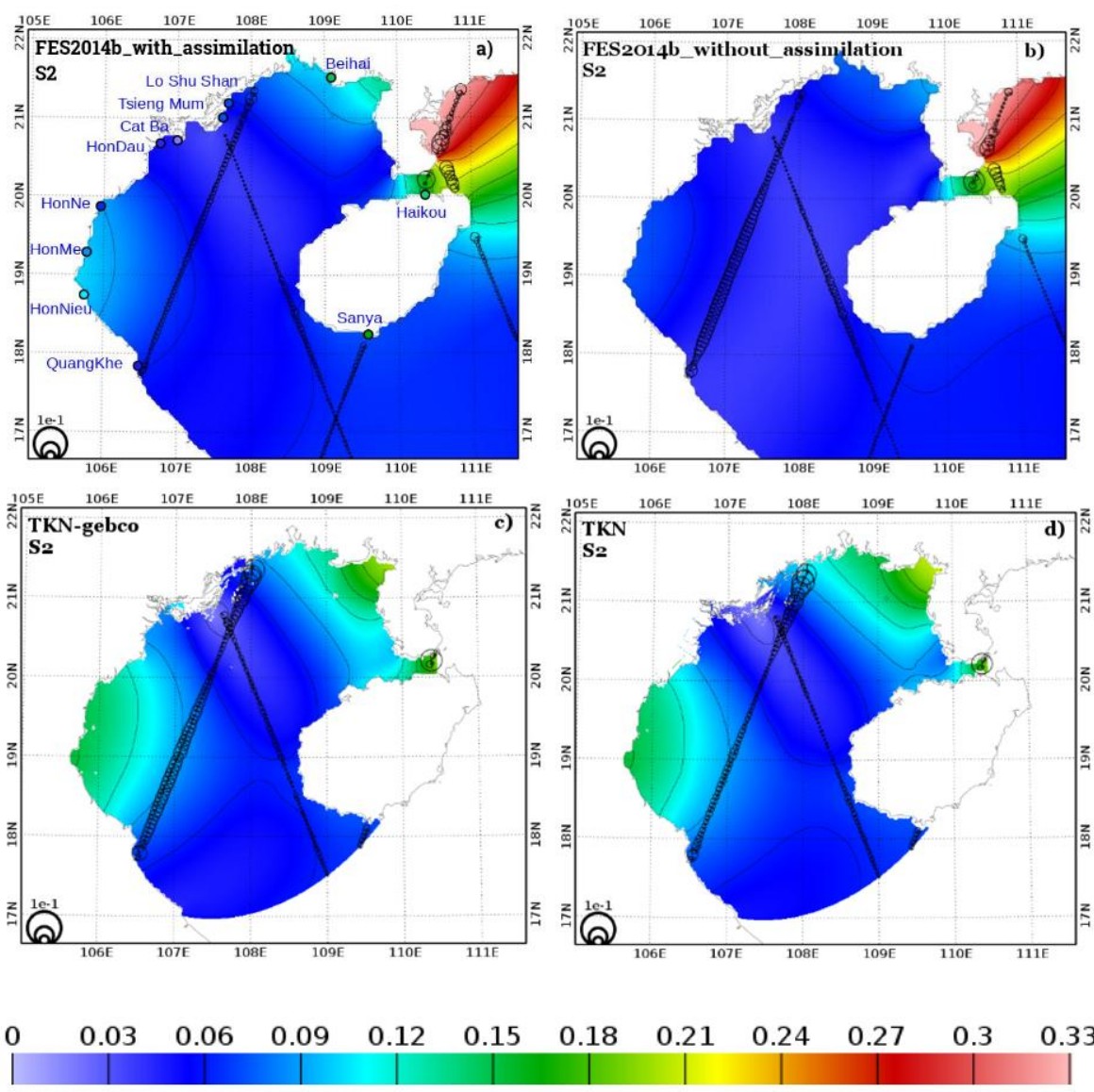

**Figure 8: Same as Fig. 5 for S2.**




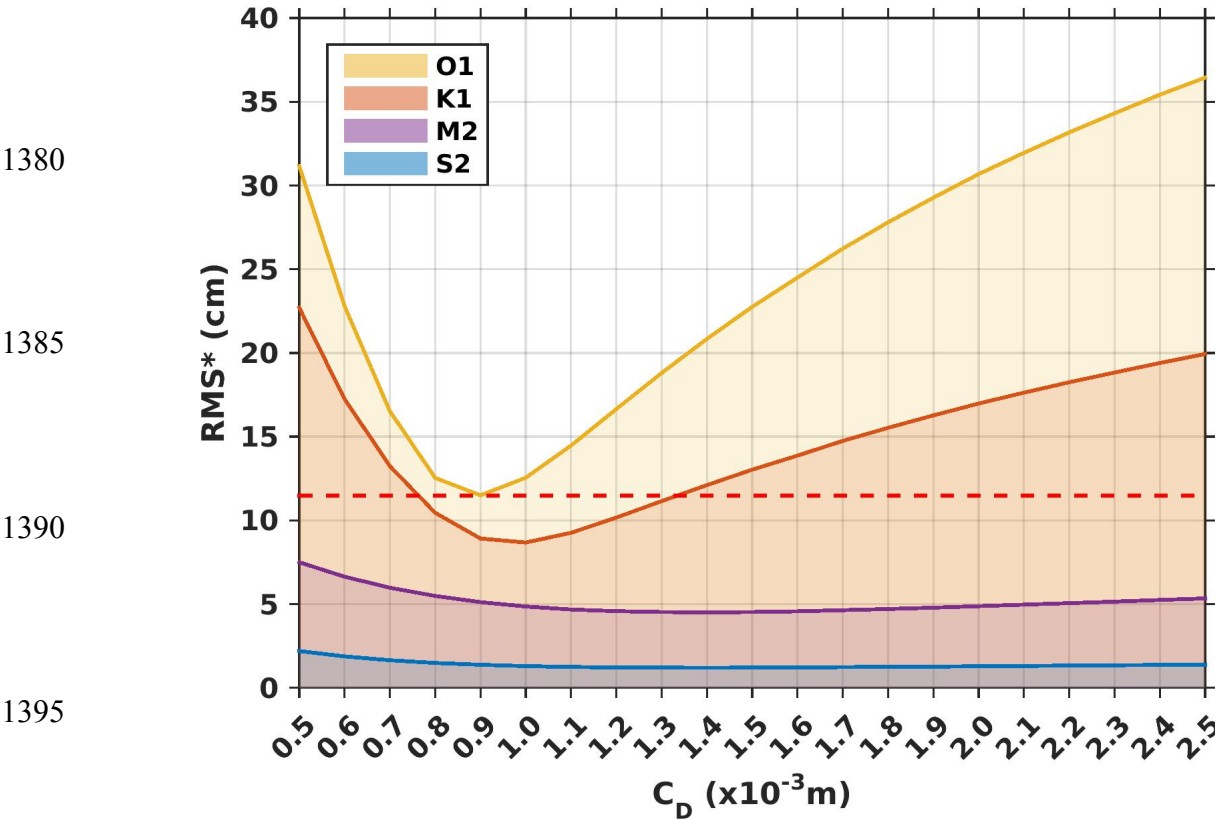

**Figure 9: Model complex errors (Appendix ; Eq. 6) relative to altimetry alongtrack data for tests performed with varying the values of the uniform drag coefficient $C_D$ over the domain (SET1). The space in between two lines corresponds to the error for each wave. The yellow line therefore corresponds to the cumulative error for all four waves. The red dashed line corresponds to the smallest cumulative error, here equals to 11.50 cm, and obtained for $C_D = 0.9 \times 10^{-3}$ m.**

1425

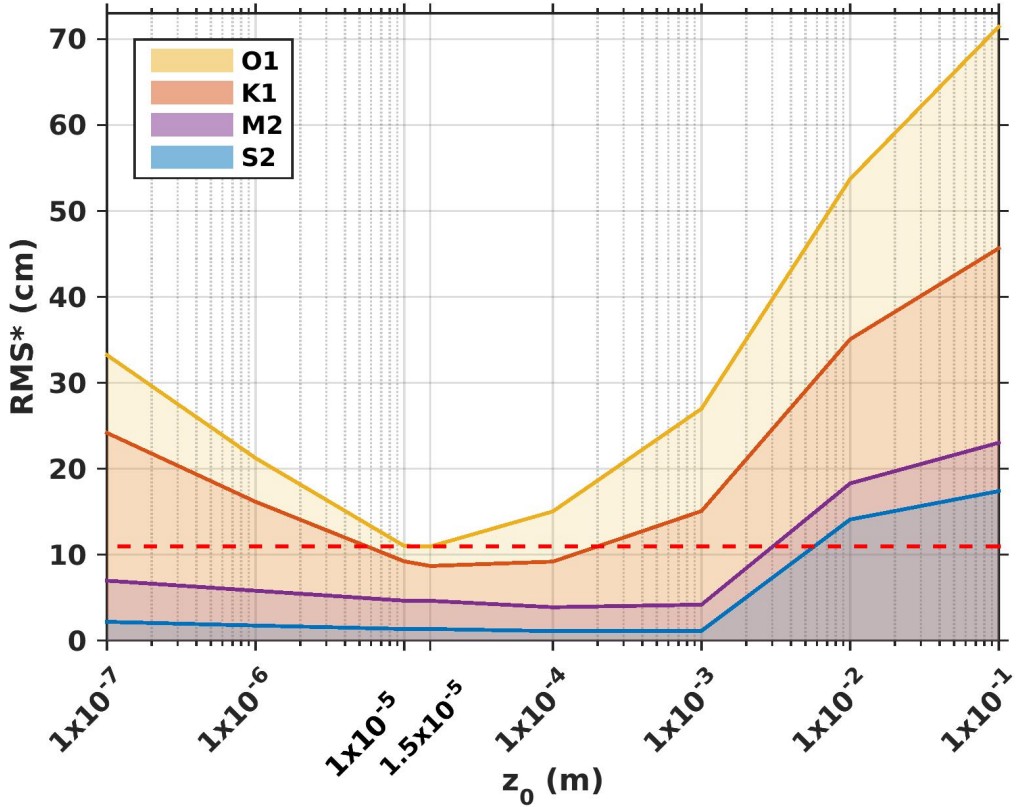

**Figure 10: Model complex errors (Appendix ; Eq. 6) relative to altimetry data for tests**
**performed with varying the values of the uniform $z_0$ over the domain (SET2). The space in between two lines corresponds to the error for each wave. The yellow line corresponds to the cumulative errors for all four waves. The red dashed line corresponds to the smallest cumulative error, here equals to 10.96 cm, and obtained for $z_0=1.5 \times 10^{-5}$ m.**





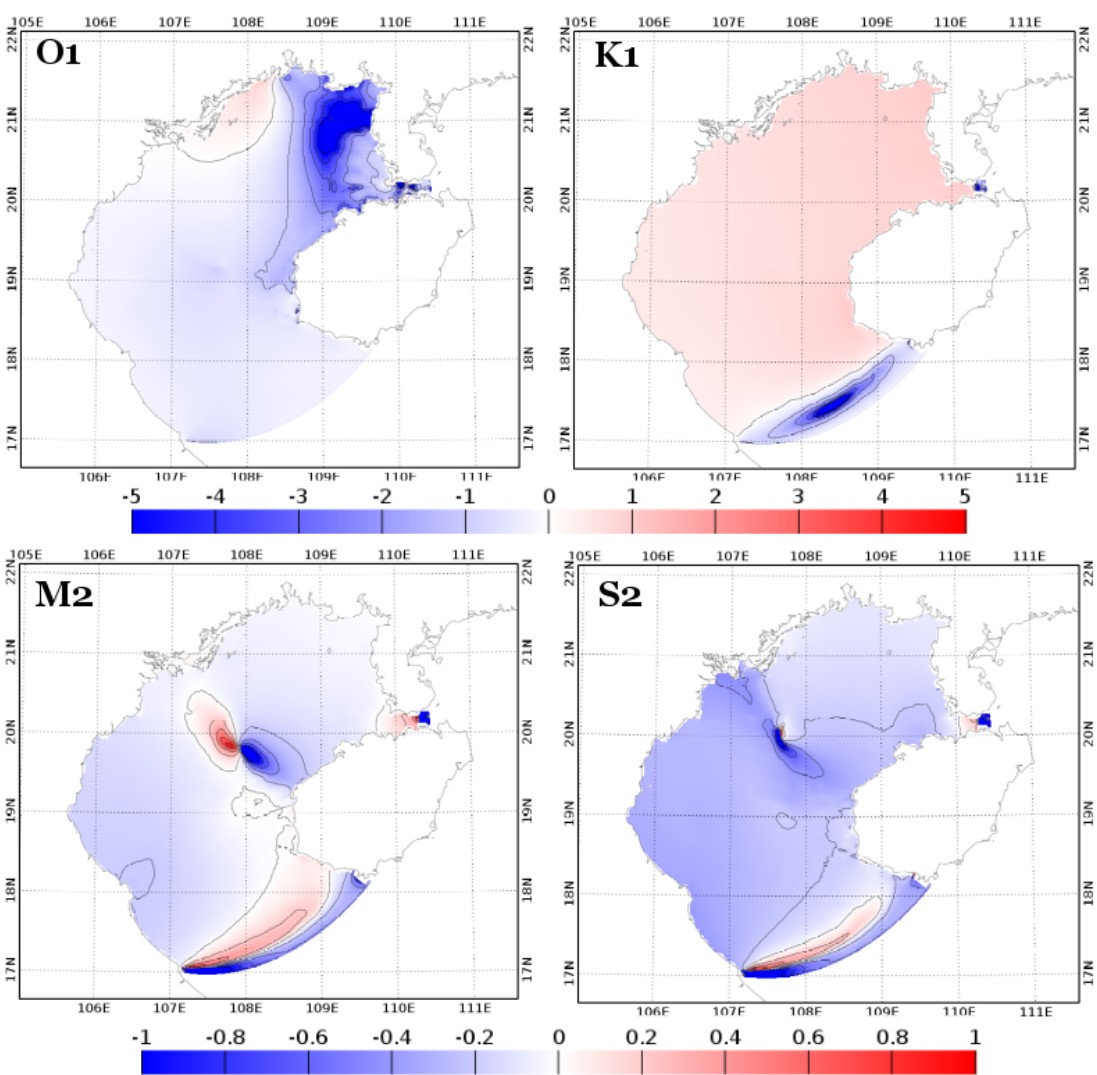


**Figure 11: Relative differences (in %) between simulation with $C_D=f(z_0=1.5e^{-5},H)$ and simulation with $C_D=0.9e^{-3}$ m compared to FES2014b-with-assimilation (as a reference) for the tidal harmonics of O1, K1, M2 and S2.**




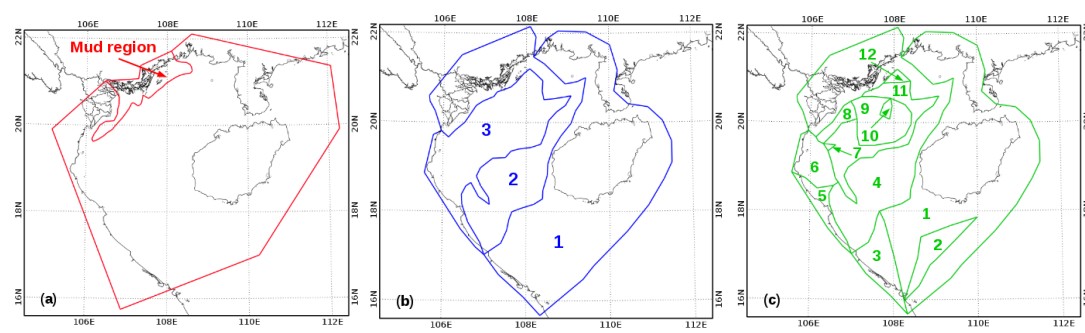

**Figure 12: Spatial partitioning of the domain for the set of experiment SET3, (a), for SET4 (b) and for SET5 (c).**



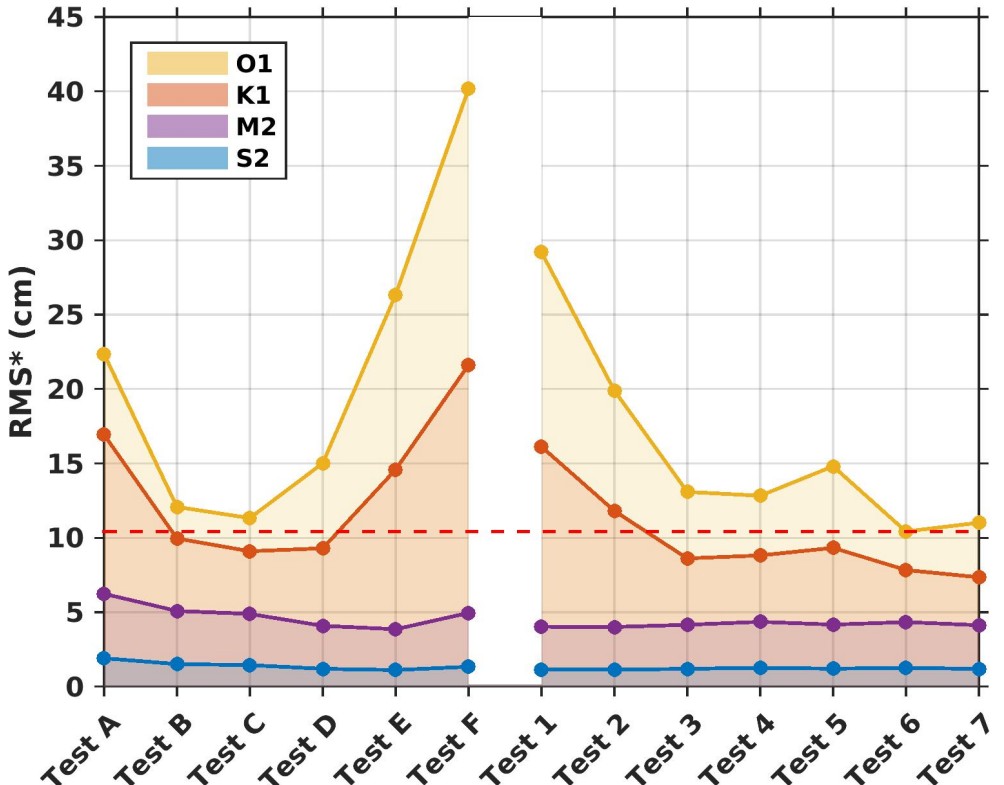

Figure

**13: Model complex errors (Appendix; Eq. 6) relative to altimetry data for tests listed in Table 1 performed with non-uniform values of $z_0$ (SET3 and SET4). The space in** 1525 **between two lines corresponds to the errors for each wave. The yellow line corresponds to the cumulative errors for all four waves. The red dashed line corresponds to the smallest cumulative error, found for Test 6 (SET 4), equals to 10.43 cm.**

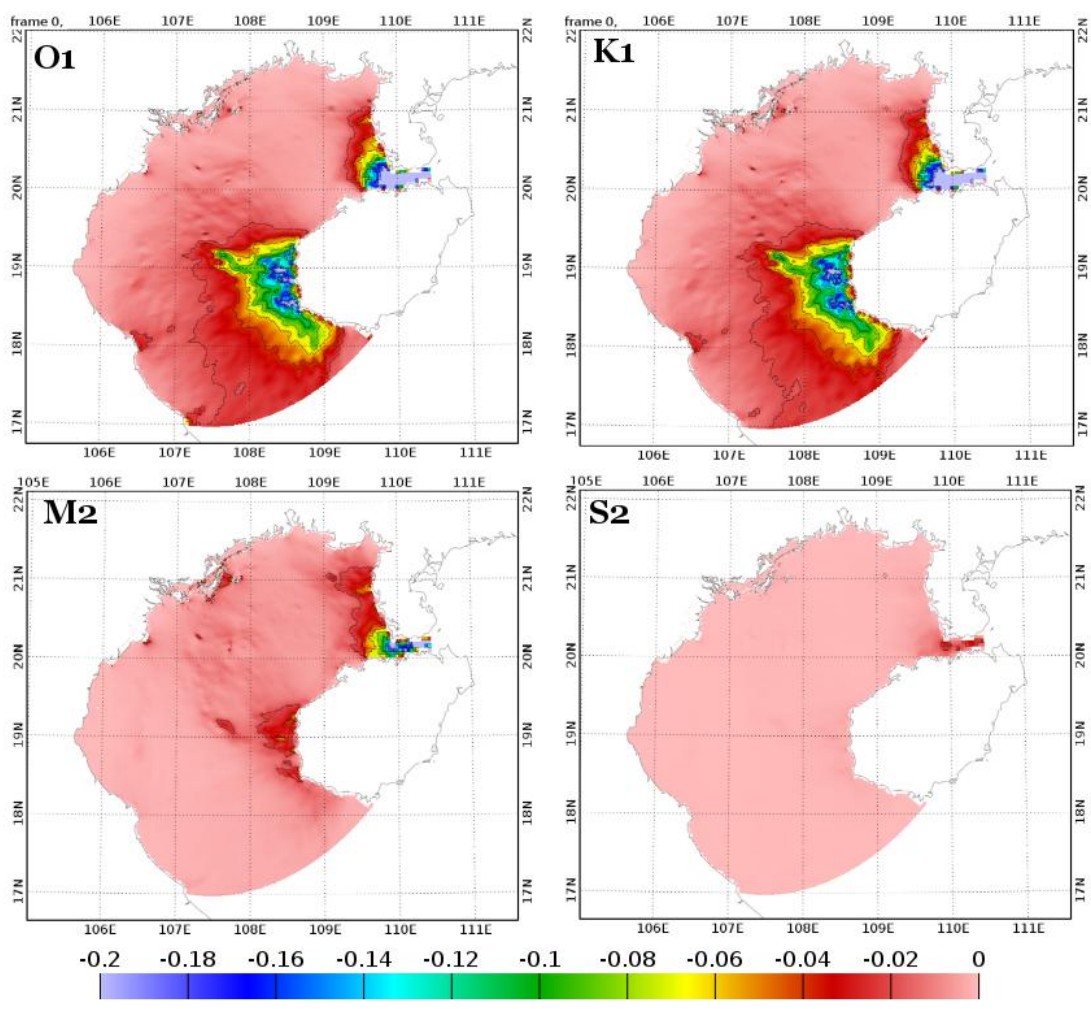

**Figure 14: Bottom dissipation flux (W m⁻²) for O1, K1, M2 and S2 computed from model outputs of simulation "TEST 6".**




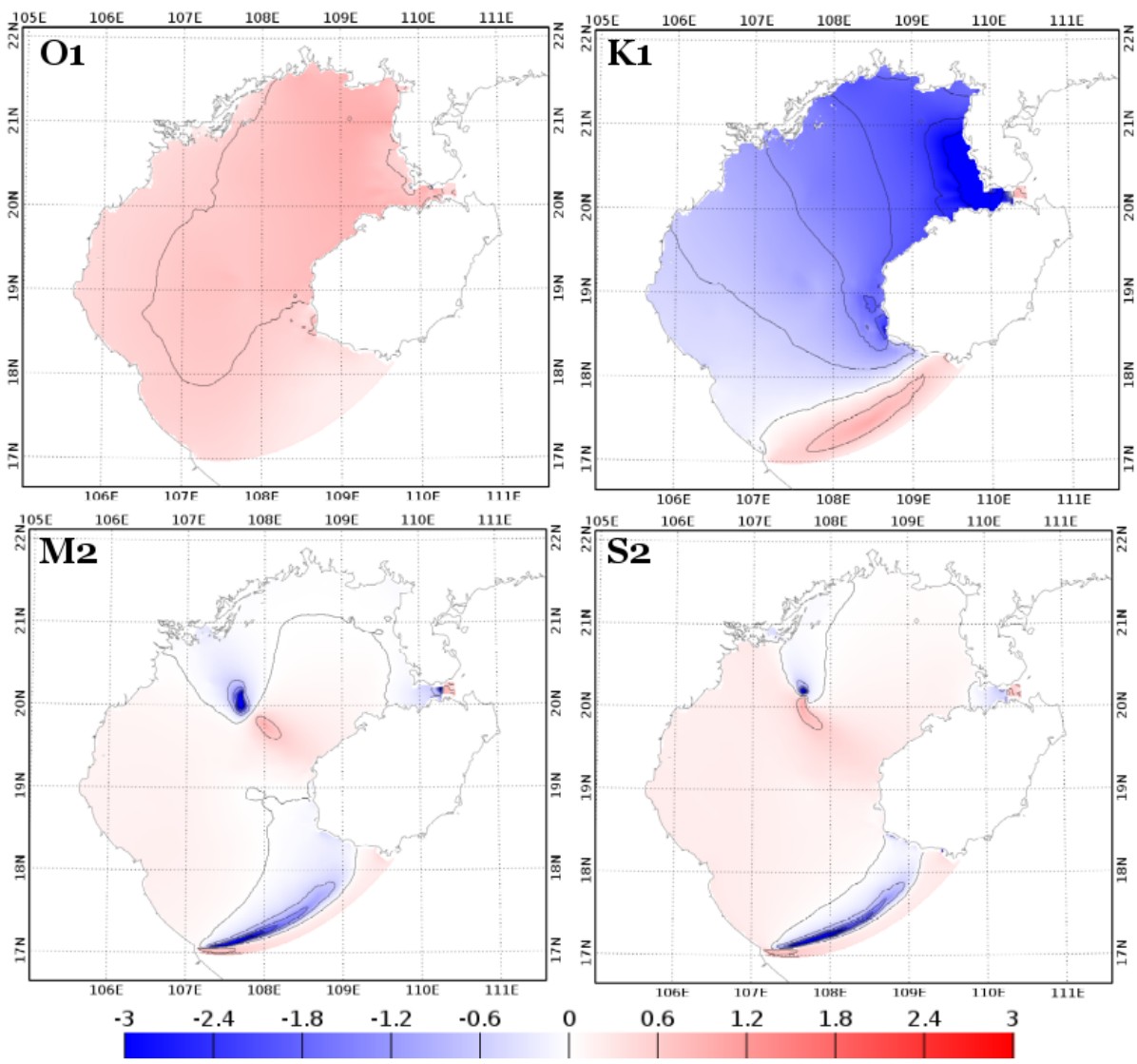

**Figure 15: Relative differences (in %) between simulation TEST6 and simulation with $C_D=f(z_0=1.5e^{-5},H)$ compared to FES2014b-with-assimilation (as a reference) for the tidal harmonics of O1, K1, M2 and S2.**

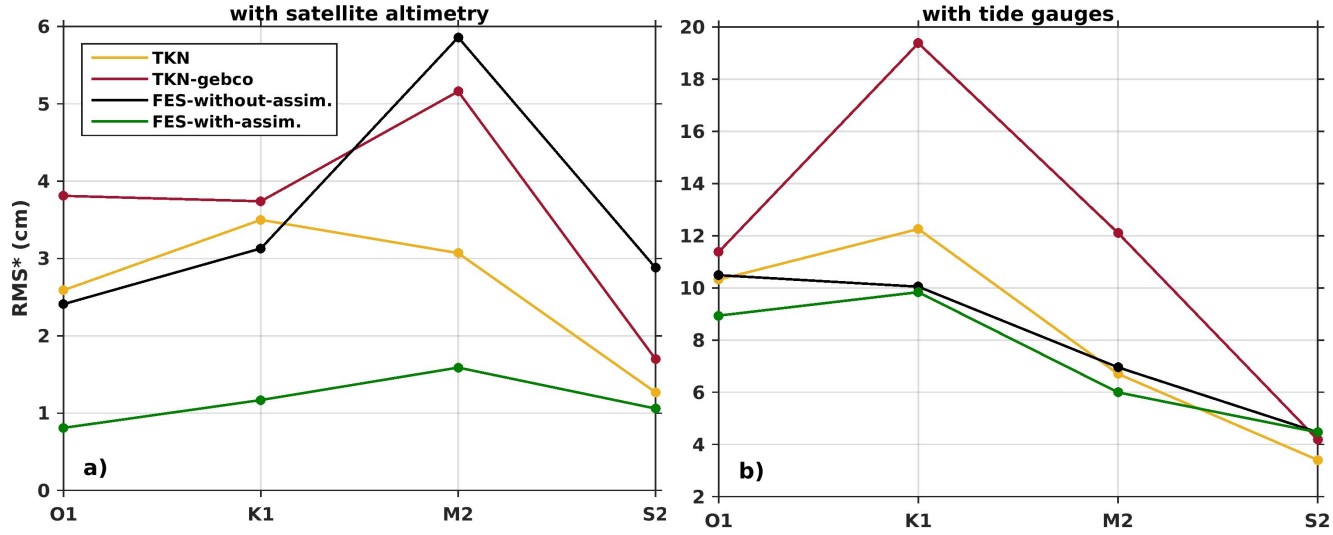

**Figure 16: RMS* errors (Appendix ; Eq. 6)**
**between numerical simulations (TKN, TKN-gebco, FES2014b-without-assimilation and**
**FES2014b-with-assimilation) and altimetry data (a) and tide gauges (b) for O1, K1, S2**
**and M2.**

| SET 3 | A | B | C | D | E | F | SET 4 | 1 | 2 | 3 | 4 | 5 | 6 | 7 |
|---|---|---|---|---|---|---|---|---|---|---|---|---|---|---|
| **Mud region (r)** | $1.18 \times 10^{-4}$ | $1.18 \times 10^{-4}$ | $1.18 \times 10^{-4}$ | $1.18 \times 10^{-4}$ | $1.18 \times 10^{-4}$ | $1.18 \times 10^{-4}$ | **Region 1 ($z_0$)** | $1.0 \times 10^{-2}$ | $1.0 \times 10^{-3}$ | $1.0 \times 10^{-4}$ | $1.0 \times 10^{-4}$ | $1.0 \times 10^{-4}$ | $1.5 \times 10^{-5}$ | $1.5 \times 10^{-5}$ |
| **$z_0$ in the rest of the domain** | $1.0 \times 10^{-6}$ | $1.0 \times 10^{-5}$ | $1.5 \times 10^{-5}$ | $1.0 \times 10^{-4}$ | $1.0 \times 10^{-3}$ | $1.0 \times 10^{-2}$ | **Region 2 ($z_0$)** | $1.5 \times 10^{-5}$ | $1.5 \times 10^{-5}$ | $1.5 \times 10^{-5}$ | $1.5 \times 10^{-5}$ | $1.0 \times 10^{-4}$ | $1.5 \times 10^{-5}$ | $1.5 \times 10^{-5}$ |
| | | | | | | | **Region 3 ($z_0$)** | $1.0 \times 10^{-4}$ | $1.0 \times 10^{-4}$ | $1.0 \times 10^{-4}$ | $1.5 \times 10^{-5}$ | $1.5 \times 10^{-5}$ | $1.0 \times 10^{-4}$ | $1.3$ |

**Table 1: Description of SET 1 and SET 2 (in m).**








| Tides | O1 | | K1 | | M2 | | S2 | |
|---|---|---|---|---|---|---|---|---|
| | Amplitude (cm) | Phase (°) | Amplitude (cm) | Phase (°) | Amplitude (cm) | Phase (°) | Amplitude (cm) | Phase (°) |
| **TKN** - this study *Compared to Satellite altimetry* | 1.5 | 3.7 | 1.9 | 5.4 | 2.3 | 7.6 | 0.9 | 14.7 |
| **FES-with-assimilation** *Compared to Satellite altimetry* | 0.8 | 3.9 | 1.1 | 4.4 | 1.5 | 3.2 | 1.0 | 7.5 |
| **FES-without-assimilation** *Compared to Satellite altimetry* | 2.6 | 9.5 | 2.6 | 6.5 | 6.0 | 10.6 | 2.5 | 13.2 |
| **Minh et al. (2014)** *Compared to Satellite altimetry* | 2.4 | 8.4 | 2.8 | 10.4 | 8.0 | 7.8 | 2.4 | 17.7 |
| **Chen et al. (2009)** *Compared to Gauge stations* | 3.0 | 9.0 | 5.4 | 8.9 | 2.3 | 6.7 | 2.8 | 22.0 |

**Table 2: Mean absolute differences (Appendix; Eq. 7) of amplitudes (in cm) and phase (in deg) of M2, S2, O1, K1 constituents between our reference TKN and satellite altimetry. For comparison, the two FES products compared with satellite altimetry, the work of Minh et al. (2014) compared with satellite altimetry and the work Chen et al., (2009) compared to gauge stations are presented.**
