# Peer review of "Violaine Piton1,2, Marine Herrmann1,2, Florent Lyard1, Patrick Marsaleix(3), Thomas Duhaut3, Damien Allain1, Sylvain Ouillon1,2"

_Geoscientific Model Development, 2019_

## Referee Comment (RC1) · Anonymous Referee #1 · 30 Aug 2019

**Comments on "Sensitivity study on the main tidal constituents of the Gulf of Tonkin by using the frequency-domain tidal solver in T-UGOm."**

**I General comments**

The objective of the paper – as presented in the introduction – is "to propose a robust and simple approach that allows to improve the tidal representation in the Gulf of Tonkin and to complement the previous studies in tidal modelling of the area." However, one of the objective is also to improve the Gulf of Tonkin configuration – bathymetry and bottom friction parameterization – using TUGO hydrodynamic code (that runs very fast), in order to use this optimized configuration for the coupled hydrodynamic/sediment model SYMPHONIE-MUSTANG (that runs more slowly). It could probably be mentionned at the beginning of the paper.

The paper presents many interesting results, particularly the sensitivity to bathymetry and bottom stress parameterization. We totally agree with the importance of the bathymetry and bottom stress in hydrodynamic modelling, and this paper is really welcome. However, the results are presented in a linear way, probably as the authors conducted the study and found them – this is particularly true for the section 2.1 Shorelines and bathymetry construction. It would be more valuable for the paper to present more analysed results and clearer conclusions. A discussion could include a reflexion around how significant is the choice of the bottom stress parameterization (e.g. drag coefficient constant Cd=cst instead of varying with z0 and H Cd=f(z0,H)).

*Bottom stress sensitivity*
The main point is that the choice of a parameterization rather than another is not justified, and finally, we do not really understand why the authors choosed to vary spatially z0 instead of Cd, as the choice of Cd constant all over the domain (SET 1) gives results very similar to Cd varying with z0 (SET2), i.e. cumulative errors are very close to each other (respectively 11.50 cm and 10.96 cm). Moreover, the fact of varying spatially the bottom friction (here z0) is not very convincing. Indeed, the choice of three areas (SET4) finally leads to results very close to one area (SET2) as two of the three areas as the same z0 than the SET2 optimized (z0 =1.5 $10^{-5}$ m) and without surprise, the cumulative errors are also very similar, even if slightly better with SET3 (10.43 cm instead of 10.96 cm for SET2). More disturbing is the fact that increasing the number of areas, the results are worse (SET5 with a cumulative error of 12.29 cm instead of 10.43 cm for SET4). This raises the question of the robustness of the method. The last point is that choosing for the mud area a linear expression instead of a quadratic one has no significative influence (SET 3). Finally, the sensitivity study allows to optimize bottom parameters (e.g. r, Cd or z0), but the sensitivity to a parameterization rather than another (linear, quadratic with Cd constant, quadratic with Cd=f(z0) with z0 constant, or even quadratic with Cd=f(z0) and z0 varying spatially) is not well demonstrated (e.g. similar cumulative errors). As a consequence, the reasons of choosing one parameterization rather than another is not clearly justified in the paper (except a cumulative error slightly lower). A discussion would be welcome to clarify this point: How significant is the choice of the parameterization between Cd=cst or Cd=f(z0) or … ? Why is it here not so significant ?

*Bathymetry sensitivity*
Results show a clear improvement between GEBCO and improved bathymetry. However, the use of bathymetry from nautical charts could reduce the depth (because charts are made for navigation purpose, see explanation in the specific comments) and lead to an overestimation of the tide. The improvement between the results and existing atlas FES2014b-hydro (without assimilation) is not so significative. For example, if we look at M2 (Figure 6), results show lower errors in the South but greater in the North, with significant differences between TKN model and FES2014bhydrodynamics. The overestimation of the amplitude nearshore could be partly due to the use of the bathymetry from the nautical charts. Note that the part of the FES2014b outside the Gulf of Tonkin could be masked, as it is not modelled in TKN. Otherwise, the scale of the figure is not appropriate, e.g. in Figure 7 for S2 the scale ranges from 0 to 0.33 m (note that there is no unit on the figures) whereas the maximum in the Gulf of Tonkin is only 0.19 m. Another point is that comparisons were made only between the model and altimetry, and comparison with **tides gauges** would be of great interest for the paper. For example, on Figure 6 authors could add dots at tide gauges locations colored with corresponding M2 amplitude. Finally, the TKN model has clearly errors greater than FES2014b-synthesis (with assimilation) which shows that the first objective (improve tidal model) is not really reached.

Maybe the paper would be clearer if the part 3.2 "Sensitivity to the bathymetry and assessment of tidal solution" could be separated into two parts, as here sensitivity to the bathymetry and assessment of tidal solutions are mixed, which is quite confusing. Results of the paper could be reorganized into three parts: 1) Sensitivity to bottom stress parameterization, that would lead to parameterization Cd=f(z0) with  z0 varying spatially 2) Sensitivity to the bathymetry, that would lead to TKN bathymetry 3) Assesment of tidal solutions, that would show tidal improvement compared with Minh et al. (2014) and Chen et al. (2009), but not so clear improvement with FES2014b-hydrodynamics (e.g. Figure 6) and clearly, no improvement compared with FES2014b-synthesis (with data assimilation). This underlines the importance of data assimilation, and the need to go on developping **satellite missions** and **in-situ campaigns**, despite the great improvements of numerical models in the last decades.

In the following, we detail the specific comments.

**II Specific comments**

- line 45: the "strong improvement (compared to pre-existing tidal atlases)" of tidal solution is not so clear compared with FES2014b-hydro (without assimilation, see general comment for Figure 6). Moreover, the model errors are really greater than FES2014b-synthesis (not surprising, as this last one is with data assimilation).
- line 101 and following: add on Figure 1 geographic elements quoted in the text as for example Gulf of Tonkin, Hainan Strait, Hainan Island, Zhanjiang Peninsula, Qinjjiang, Nanliu, Yingzai rivers, Hai Phong harbour…
- line 153 and following: precise the expression of the tidal form factor, and the existence of four regimes.
- lines 153-162: maybe a map of F would be useful, as the variation of F is here described spatially.
- line 181: the objectives are not clear. It is mentionned to improve the tidal representation, whereas the TKN tidal model is finally not improved compared to FES2014b-synthesis. Probably introduce here as an objective the idea of using the tide as a response to calibrate the bottom friction parameterization. Objectives are also to improve model configuration (bathymetry and bottom friction) with TUGO (running fast) with the final goal to run the configuration on  SYMPHONIE-MUSTANG (that runs more slowly).
- line 189: "in poorly sampled regions", precise in terms of what, bathymetry? sea level? tidal currents? tide gauges?
- line 193: "in situ data and soundings are consequently rare and extremely valuable", is it possible to list these data in the area? Particularly tide gauges may be of great interest for this study.
- line 230: why did you choose Bing as "the reference"? Is there is a paper reference for this?
- line 235: "OpenStreetMap shoreline is most of the time shifted", is there is an explanation for that?

- line 236: "The GSHHG dataset suffers from the same problem but shifted by up to 500m eastwards.", the shift is here very significant, is there is an explanation for that?

- line 238: "matching the reality", what is considered as the "reality", and why? Bing maps?

- line 240: shorelines from POCViP, is there is a reference paper for this software? What are the data behind this software?

- line 257: the GECBCO resolution is of approximately 1 km. There is an important difference between the grid resolution and the data resolution, as the data resolution could be lower than the grid resolution (and interpolated). Of which resolution are we talking here? Grid resolution ? Do we know what is the data resolution ?

- line 261: "digitalized nautical charts" Charts are made for navigation, and near the coast, the hydrographer generally chooses the lower soundings (shallow waters) for security purpose. As a consequence, the bathymetry from nautical charts in coastal areas gives shallower waters than "real" bathymetry. This should be mentionned, as it could partially explain the overestimation of tidal amplitude with TKN compared with FES2014b-hydrodynamics.

- line 285: "TONKIN_bathymetry dataset is not considered as the truth", rather say that due to sampling, there are still uncertainties on the bathymetry. Also mention problems linked with nautical charts (shallower waters, see comment above).

- line 300 and following, no reference of TUGO for "storm surges simulations"?

- line 319: version of the code is a little bit complicated… "2616:78a276dd7882 of 2018-07-22 320 13:17 +0200"

- line 320: TKN is the name of the code or of the configuration? Not clear here.

- line 332: "The quadratic parameterization may be obsolete and a linear parameterization more adequate". Is there is a justification or a reference for this sentence? The results will show the contrary.

- line 340: the "final goal", i.e. hydrodynamic-sediment transport with SYMPHONIE-MUSTANG could be introduced earlier.

- line 377: the names FES2014b-hydrodynamics and FES2014b-synthesis are not really explicit. Choose for example FES2014b-without-assimilation and FES2014b-with-assimilation or something else, but more explicit.

- line 383: is there is a reference for FES2014b?

- line 402 and following: 2.3.1.1 Bottom stress parameterization, this section is not clear enough. The three parameterizations (2) (3) (4) are finally two paramaterizations (1a) and (1b), the first one with Cd=constant or Cd=f(z0,H) and the second one with r=constant. Particularly, the sentence 421 is not clear "In this study, we test three commonly used parameterizations: a constant drag coefficient Cd assuming a constant speed profile or a linear speed profile, and a drag coefficient Cd depending upon the roughness height z0".  This paragraph could be rewritten to be clearer.

- line 447: "In presence of fluid mud,..." repetition, yet said before, line 415

- line 457: (2) and (4) are linked with (1a) parameterization, whereas (3) is linked with (1b) parameterization. The way the parameterizations are presented is confusing.

- line 464: "two of the parameterizations described above: a quadratic bottom stress with a uniform drag coefficient Cd (Eqs. 1a and 2) and a logarithmic variation of Cd depending on a uniform bottom roughness height z0 (Eq. 4)" is not very clear. It would be more appropriate to talk about a quadratic bottom stress with a drag coefficient constant (Cd=cst) or varying with the roughness length (Cd=f(z0,H)).

- lines 521: "Spatially varying uniform friction parameters induce the best results on the tidal solutions rather than uniform parameters." Is the improvement significative? Moreover, 12 areas give worse results than three areas, and three areas correspond finally to only two. Is the method robust enough ?

- line 522: "However, prescribing a linear parameterization in supposed fluid mud areas does not allow to significantly improve the solutions" How to explain this?

- line 530 and following "3.1.1 Sensitivity to the value of spatially uniform parameters"

To be clearer, this title could be "Sensitivity to a constant or varying Cd", because SET1 corresponds to Cd=cst, and SET2 corresponds to Cd=f($z_0$,H). Optimisation conducts to Cd=0.9 $10^{-3}$ m, and $z_0$=1.5 $10^{-5}$ m. A map of Cd for $z_0$=1.5 $10^{-5}$ m would help to see the differences in term of Cd between the two parameterizations. The cumulative errors between Cd=cst or varing with $z_0$ are very similar (11.5 and 10.96 cm). Is there is a significative improvement with Cd=f($z_0$,H)? It is not justified why the authors chose Cd=f($z_0$) rather than Cd=cst for the following. Indeed, SET4 could have been made with Cd varying spatially, instead of $z_0$ varying spatially.

- line 585 and following: "Sensitivity to the value of spatially varying roughness length (SET3, SET4, SET5)"

This part could be separate in two parts "Sensitivity to a quadratic or linear stress" (which correspond to SET3 compared to SET2) and "Sensitivity to a roughness length varying spatially" (which corresponds to SET4 and SET5 compared to SET2).

SET3 has been conducted with a constant value of r and varying value of $z_0$. Without surprise, optimization leads to $z_0$=1.5 $10^{-5}$ m, which is the optimized value of SET2. Why not fixing the $z_0$ value ($z_0$=1.5 $10^{-5}$ m) and make vary the r value? This could lead to an optimized r value, probably different from 1.18 $10^{-4}$ m from Le Bars et al. (2010), and perhaps results would show more sensitivity to a quadratic or linear stress (depending on the r value).

SET4: the three areas are finally only two areas (Regions 1 and 2 with the same $z_0$), and the $z_0$ in areas 1 and 2 is the one corresponding to SET2 optimized ($z_0$=1.5 $10^{-5}$ m). Finally, SET4 is not so different from SET2, with very similar cumulative error (10.43 cm instead of 10.96 cm). Is the improvement significative?

SET5: how to explain that it is worse with 12 areas? Why don't we converge also to $z_0$=1.5 $10^{-5}$ m? It could be interesting to include these results and analyse them, otherwise, we don't have enough elements to understand, and we can wonder if the method is robust enough.

- line 622: "3.2 Sensitivity to the bathymetry and assessment of tidal solution", this section could be split in two parts, see comment in general comments.

- line 649: it is not so clear why TKN and TKN-gebco show bigger errors than FES2014b-hydrodynamics. We understand than K1 is less sensitive to bathymetric variations, but this is probably not the only explanation.

- line 665: TKN is better than Minh et al. (2014) and Chen et al. (2009). Why? This could be explained.

- line 686 and following: the acronym SLA is not detailed, are we talking about Sea Level Anomaly? This term is generally corrected from tide. It is not clear here.

- line 756 and following: the fact that TUGO is used to prepare a best configuration for SYMPHONIE-MUSTANG could appear earlier in the paper (e.g. in the objectives). Otherwise, it is not clear if the objective of the paper is to improve the tidal model or improve the configuration (bathymetry) and parameterization (bottom stress) for SYMPHONIE-MUSTANG.

- line 763: it is clear that the new bathymetry improves the results compared with GEBCO, but it is not clear if the final configuration TKN is improved compared with FES2014b.

- line 784: "the use of a constant Cd parameterization or the use of a Cd depending on the roughness length led to **fairly similar results**" We totally agree with this conclusion, that could appear earlier in the paper (in the results). As a consequence, why choose a Cd depending on the roughness length instead of a constant Cd for SET3/SET4/SET5?

- line 791: "the regionalisation of the roughness length into three regions, for addressing the issue of representing the complexity of seabed composition and morphology, **moderately improved the accuracy of our simulation**, with a lowest cumulative error for all four waves of 10.43 cm", instead of 10.96 cm. We totally agree with this conclusion, is the improvement significative enough?

- line 794: "Finer local adjustments of the roughness length or the choice of a linear velocity profile in the area of fine mud, **did not improve** the accuracy of our simulations." We totally agree with this conclusion, how to explain that there is no improvement ?
Finally all the SETs - once optimized individually - are very close to each other, in term of cumulative error.
- line 800 : "results therefore quantitatively showed the importance of the bathymetry and shoreline dataset and of the choice of bottom friction parameters for the representation of tidal simulations over a shallow area like the GoT". This could be clarified. The results show that the choice of the bottom stress parameterization is not so important (e.g. SET2 optimized with Cd=cst gives similar results than SET3 optimized with Cd=f(z0,H), in terms of cumulative error), but the value of the bottom parameter (e.g. Cd=0.9 10-3 m for SET1 or z0=1.5 10-5m for SET2) is important, as it impacts clearly the cumulative error (e.g. Figure 8).
- line 806: "Our resulting configuration brought a clear improvement in the tidal solutions compared to previous 3D simulations from the literature and to the tidal atlas FES2014b (without data assimilation) for the semi-diurnal waves." The improvement compared to FES2014b is not so clear if we look at Figure 6 for example. The addition of tide gauges data should greatly help to qualify the results.
- line 813:  "Using bathymetry data available from digitalized navigation charts was a relatively simple way (compared to performing additional in-situ measurements) to significantly improve the representation of topography in the coastal and estuarine areas of the GoT". We agree, however, as mentionned above, bathymetry could be underestimated (shallower waters) because charts are made for navigation and the shorter soundings are choosen for security reasons. The use of nautical charts could then lead to an overestimation of the tidal amplitude in some coastal areas.
- Table 2, FES2014b-hydrodynamics and FES2014b-synthesis are clearly missing.

---

## Referee Comment (RC2) · Anonymous Referee #2 · 20 Sep 2019

General comments:

The authors applied the frequency-domain tidal solver in the hydrodynamic unstructured grid model T-UGOm to examine the sensitivity of the main tidal constituents of the Gulf of Tonkin. The model results are compared with observation s collected from satellite. The model validation suggests that the model is able to capture the tidal dynamics in the Gulf of Tonkin. The authors also constructed a series of sensitivity model experiments to test the bathymetry and bottom friction parameterization. In my opinion, the paper is potentially a valuable contribution to the scientific literature of the Gulf of Tonkin, as the model constructed by the authors is able to well capture the tidal

dynamics in the Gulf of Tonkin. More over the paper is clear and well written. In general, the figures are neat. I recommend publication of the paper in Geoscientific Model Development Discussions after minor revisions, in response to the following concerns:

Specific comments:

1) L359-360: For the tidal open boundary condition, nine tidal constituents were considered. Why do you include the shallow water constituent M4? Does this tidal constituent contribute significantly to the tide in the GOT? How about other shallow water constituents such as MS4 and M6?

2) The model simulated tidal constants are compared with satellite data. Have the authors tried to compare the model results with the observations from tide gauge stations along the coast of the Gulf of Tonkin? How about the tidal current in the simulations? Have the authors validated the model-simulated tidal current with observations?

3) The Red River is the most important freshwater discharge in the Gulf of Tonkin. The freshwater from the Red River may influence the tide near the estuary. Have the authors considered the effect of the freshwater discharge on the tidal simuations?

Technical corrections:

L106: "Quiongzhou Strait"should be Qiongzhou Strait.
* * *

---

## Author Comment (AC1) · 22 Nov 2019

We do thank Referee#2 for his/her careful reading of our manuscript and relevant comments. Below are his/her comments (in bold), followed by our responses and description (in italics). Changes in the revised manuscript are highlighted in red.

General comments: The authors applied the frequency-domain tidal solver in the hydrodynamic unstructured grid model T-UGOm to examine the sensitivity of the main tidal constituents of the Gulf of Tonkin. The model results are compared with observations collected from satellite. The model validation suggests that the model is able to capture the tidal dynamics in the Gulf of Tonkin. The authors also constructed a series of sensitivity model experiments to test the bathymetry and bottom friction parameterization. In my opinion, the paper is potentially a valuable contribution to the scientific literature of the Gulf of Tonkin, as the model constructed by the authors is able to well capture the tidal dynamics in the Gulf of Tonkin. More over the paper is clear and well written. In general, the figures are neat. I recommend publication of the paper in Geoscientific Model Development Discussions after minor revisions, in response to the following concernsÂă:

Specific comments:

1) L359-360: For the tidal open boundary condition, nine tidal constituents were considered. Why do you include the shallow water constituent M4? Does this tidal constituent contribute significantly to the tide in the GOT? How about other shallow water constituents such as MS4 and M6?

We thank the reviewer for addressing this issue and we understand his/her concern. First of all, the main objective of our study was to calibrate the astronomical spectrum of tide as it is dominant in the GoT over the linear spectrum (Wyrtki, 1961). Therefore, 8 out of the 9 constituents simulated are astronomical constituents, while the last one (M4) is a linear tidal constituent and was chosen as a representative of all linear interactions. We agree we could have simulated MS4 instead of M4 as their patterns of amplitudes are very similar (Fig. 1 a,b). M4 and MS4 amplitudes are maximum in the northern GoT ($\sim$0.02 m), along the coast of Vietnam and at the western entrance of the Hainan Strait. These amplitudes are, however, roughly 50 times smaller than the maximal amplitudes of O1 and K1 and 15 to 35 times smaller than the amplitudes of S2 and M2, respectively (Fig. 1 a, b). Therefore, we believe simulating both M4 and MS4 or MS4 instead of M4 would not induce significant changes in the final tidal solutions. Lastly, M6 amplitudes were much smaller than M4 and MS4 all over the GoT, and was therefore neglected in our simulations (Fig. 1 c), as again, it should not induce

[Figure]

significant changes in the final tidal solutions.

Figure 1: Tidal amplitudes (m) of M4, MS4 and M6 from FES2014b-with-assimilation product.

–>Following this comment, we have added a brief explanation about this choice of tidal constituents in the revised version of the manuscript (lines 379-381).

2) The model simulated tidal constants are compared with satellite data. a) Have the authors tried to compare the model results with the observations from tide gauge stations along the coast of the Gulf of Tonkin? b) How about the tidal current in the simulations? Have the authors validated the model -simulated tidal currents with observations?

a) We thank the reviewer for suggesting comparison of our simulations with tide gauge data, which contributes to make our model evaluation more robust. Following this comment we compared our results to tidal harmonics of 11 stations located along the Gulf coasts (see locations and names in Fig. 2 and on Fig. 4 a of the revised manuscript). These data are provided by the International Hydrographic Organization (https://www.iho.int/) and are available upon request at https://www.admiralty.co.uk/ukho/tidal-harmonics. –>This dataset is detailed in the manuscript lines 537-542.

Figure 2: O1 tidal amplitude (in m) from FES2014b-with-assimilation superimposed with locations of tide gauges (black cross). This figure now corresponds to Fig. 5 a of the revised manuscript.

RMS* errors between modelled and observed tidal harmonics from tide gauges are now shown in Fig. 16 b of the manuscript (and on Fig. 3 of this document). First, compared to TKN-gebco, TKN gives smaller errors for all four waves considered: RMS* for K1 are reduced by ∼40% in TKN compared to TKN-gebco and RMS* for M2 are reduced by ∼45% in TKN compared to TKN-gebco. This result again confirms that

[Figure]

the use of the improved bathymetry dataset significantly improves the tidal representation over the GoT. Second, TKN configuration shows also smaller RMS* errors than FES2014b-without-assimilation simulation for O1, M2 and S2. In addition, TKN even minimizes S2 RMS* errors compared to FES2014b-with-assimilation. Our improved configuration however fails to improve the solution of K1, compared to the two FES2014b products.

Figure 3: RMS* errors between numerical simulations (TKN, TKN-gebco, FES2014b-without-assimilation, FES2014b-with-assimulation) and tide gauges harmonics for O1, K1, M2 and S2.

These results suggest that TKN configuration brings a clear improvement in tidal solutions compared to TKN-gebco configuration, and a slight improvement compared to FES-without -assimilation. They also confirm that FES-with-assimilation logically produces the smallest error, thanks to the use of assimilation and of a unstructured grid specially designed to represent the complexity of coastline and coastal bathymetry. –>This is now further detailed and discussed in the manuscript lines 7435759.

b) Regarding the reviewer's concern about modelled tidal currents, we indeed did not evaluate our model against in situ observation. Until recently, current-meter observations in the GoT were scarce (only limited to specific areas such as the Hainan Strait) and limited in time (daily to seasonal). Deriving clean and robust tidal currents from these datasets would have been challenging, if not impossible, as it requires long and accurate time-series. Since late 2012 however, hourly surface currents data are available from HF radars, at a resolution of 5.85 km. These HF radars are part of the Global High Frequency Radar Network (Roarty et al., 2019). We soon expect to derive tidal currents from these valuable data. From now however, the dataset still suffers from correction errors and varying spatial coverage due to seasonal monsoon patterns. TUGO has already proven its accuracy in reproducing shelf tidal currents, for example on the shelves around Australia (https://www.researchgate.net/publication/322331188_Assessment_of_the_FES2014_Tidal_Currents_on_the_shelves_ar
), it therefore certainly reproduces tidal currents over the Gulf of Tonkin shelf. However, we completely agree that this assumption will have to be confirmed thanks to future comparison with quality checked HF radar data (Rogowski et al., 2019). This will be done in the on-going phase of our work, i.e. the implementation and evaluation of the 3D SYMPHONIE-MUSTANG model used for our study of dynamics and sediment transport variability in the Gulf of Tonkin.

–>Following this comment, we acknowledged this issue in the revised version of the manuscript lines 320-321 and lines 951-954.

3) The Red River is the most important freshwater discharge in the Gulf of Tonkin. The freshwater from the Red River may influence the tide near the estuary. Have the authors considered the effect of the freshwater discharge on the tidal simulations?

We understand the reviewer's concern about the potential effects of strong discharge, which is indeed strong in the region, on tide. Previous studies in the Red River estuaries and plume suggest that water discharges could have an influence on tide, but rather the other way around: tides can influence water discharges. Lefebvre et al. (2012) and Vinh et al. (2018) showed that during the early wet season, spring tides enabled saline water intrusion up to 20-30 km along the Cam-Bach Dang estuary. In the Van Uc river (3rd biggest river of the Red River system), Piton et al. (under review) suggested that during the dry season, neap and spring tides were able to reverse the river flow up to 20 km upstream from the river mouth, and that spring tides at high tides were able to reverse the intense river flow of the wet season. Therefore, the potential effects of discharges on tide might only happen during the wet season at neap tides, and could be localized in the very near coastal area. Furthermore, taking the effects of water discharges on tide would not affect the statistics presented in our manuscript, as altimetry data, that we use for model performance assessments, are not available in the coastal and shallow area of the Red River Delta. Such assumptions should however be verified with a sequential model, more adapted than a spectral model to take into consideration the seasonal variability of water discharge which is very strong in

the region. This could be done using the 3D SYMPHONIE model implemented for our hydro-sedimentary study over the Gulf of Tonkin, and will definitely be one of our future research topic.

–>Following this comment, we acknowledge this issue lines 977-979.

Technical corrections:

L106:Quiongzhou Strait should be Qiongzhou Strait.

We thank the reviewer for spotting this mistake, is has been changed accordingly line 107.

Please also note the supplement to this comment:
https://www.geosci-model-dev-discuss.net/gmd-2019-40/gmd-2019-40-AC1-supplement.pdf

———————————————

[Figure]

[Figure]

**Fig. 1.**

[Figure]

**Fig. 2.**

[Figure]

**Fig. 3.**

---

## Author Comment (AC2) · 22 Nov 2019

We do thank Referee#1 for his/her careful reading of our manuscript and relevant comments. Below are his/her comments (in bold), followed by our responses and description of related changes in manuscript (in italics). Changes made in the manuscript are highlighted in red in the revised version of the manuscript As some remarks of the General comments are also addressed in more details in the Specific comments, we took the liberty to re organize the document by gathering together the comments tackling the same issue, in order to make our answer clearer. We hope this will ease the

reviewer's reading.

I- General comments

General comment #1 The objective of the paper – as presented in the introduction – is
'Ăăto propose a robust and simple approach that allows to improve the tidal represen-
tation in the Gulf of Tonkin and to complement the previous studies in tidal modelling
of the area.Ăă' However, one of the objective is also to improve the Gulf of Tonkin
configuration – bathymetry and bottom friction parameterization – using TUGO hydro-
dynamic code (that runs very fast), in order to use this optimized configuration for the
coupled hydrodynamic/sediment model SYMPHONIE-MUSTANG (that runs slowly). It
could probably be mentioned at the beginning of the paper.

Since the question of the clarity of the objectives addressed in this General comment
**1 is also addressed in specific comments #5 and #35, we regroup here the answers**
to those related comments to avoid repetitions make our answer clearer.

**5- line 181: the objectives are not clear. It is mentionned to improve the tidal repre-**
sentation, whereas the TKN tidal model is finally not improved compared to FES2014b-
synthesis. Probably introduce here as an objective the idea of using the tide as a re-
sponse to calibrate the bottom friction parameterization. Objectives are also to improve
model configuration (bathymetry and bottom friction) with TUGO (running fast) with the
final goal to run the configuration on SYMPHONIE-MUSTANG (that runs slowly).

**35- line 756 and following: the fact that TUGO is used to prepare a best configu-**
ration for SYMPHONIE-MUSTANG could appear earlier in the paper (e.g. in the ob-
jectives). Otherwise, it is not clear if the objective of the paper is to improve the tidal
model or improve the configuration (bathymetry) and parameterization (bottom stress)
for SYMPHONIE-MUSTANG.

We thank the reviewer for these suggestions that helped to better highlight the aim
of our work. The final aim of our work is indeed to implement an optimized configu-

ration (bathymetry, parameterizations ...) of a 3D structured grid model to study the hydrosedimentary processes and variability in the Gulf of Tonkin, and not to produce a new tidal atlas, which is the goal of state of the art products like FES2014b which are, based on the use of assimilation and a dedicated finite element grid taking into account the complexity of coastal areas. –> Following this comment, the part dealing with the description of the paper's objectives has been developed and it now integrates the objective of improving the model configuration in the frame of a hydro-sedimentary numerical study, lines 183-192. This is also reminded in the conclusions, lines 880-881.

General comment #2 The paper presents mainly interesting results, particularly the sensitivity to bathymetry and bottom stress parameterization. We totally agree with the importance of the bathymetry and bottom stress in hydrodynamic modelling, and this paper is really welcome. However, the results are presented in a linear way, probably as the authors conducted the study and found them – this is particularly true for the section 2.1 Shorelines and bathymetry construction. It would be more valuable for the paper to present more analyzed results and clearer conclusion. A discussion could include a reflexion around how significant is the choice of the bottom stress parameterization (e.g. drag coefficient constant Cd=cst instead of varying z0 and H Cd=f(z0,H)).

We thank the reviewer for thinking that our paper Âńis really welcomeÂż. Regarding the paper's construction, we agree that some work was necessary to better emphasize the main results and messages of this study. Thanks to the reviewer's suggestions and comments that were proposed throughout the General comments and the Specific comments below, we modified our manuscript taking those comments into account. One of the main concerns of the reviewer is the significance of the choice of the bottom stress parameterization, that we address in the revised manuscript in a developed discussion concerning this question, based on a synthetic figure quantifying the significance of the difference of performance between our simulations (Figure 2 and 3). This issue is discussed into details in the answer to General comment #3 and specific comments #19, 27, 28, 29, 30 and 38 below. We sincerely believe our manuscript is

now improved in terms of results analysis and conclusion. We present below in details the modifications we have made to the manuscript to improve its clarity.

General comment #3 Part 1- The main point is that the choice of a parameterization rather than another is not justified, and finally, we do not really understand why the authors choosed to vary spatially z0 instead of Cd, as the choice of Cd constant all over the domain (SET1) gives results very similar to Cd varying with z0 (SET2), I.e, cumulative errors are very close to each other (respectively 11.50cm and 10.96cm). This general comment regards the question of the choice of a parameterization based on a constant Cd vs. a varying Cd=f(z0). Since this question is also addressed in #27-A and #29, we regroup here the answers to those related comments to avoid repetitions make our answer clearer.

**29- A- line 530 and following ÂńÂă3.1.1 Sensitivity to the value of spatially uniform pa-rametersÂăÂż To be clearer this title could be ÂńÂăSensitivity to a constant or varying CD", because SET1 corresponds to Cd=cst, and SET2 corresponds to Cd=f(z0,H).**

–>We agree with this suggestion and changed the section's title accordingly line 563.

**29 - B - Optimization conducts to Cd=0.9 10-3 m, and z0=1.5 10-5 m. A map of Cd for z0=1.5 10-5 m would help to see the differences in term of Cd between the two parameterizations.**

We thank the reviewer for his/her suggestion, and plotted a map of CD=f(z0,H) with z0=1.5 10-5 m (Fig. 1). This figure allows to spot the differences between the different parameterizations: a constant CD=0.9 10-3 m or CD=f(z0,H). Values of CD range from 0.8 to 1.1 10-3 m with lower values in the deepest part of the basin and higher values along the coasts. As CD spatial variability is therefore rather weak, it also explains why both parameterizations (CD constant or CD varying with z0) produce similar results.

–> Following this comment and since the spatial variability of CD is very weak, we did not include the map for the sake of conciseness, but acknowledged this point about the

weak spatial variability of CD in the revised version of the manuscript, lines 594-598.

Figure 1: Map of CD=f(z0,H) (in m) with z0=1.5x10-5 m.

**29-C- The cumulative errors between Cd=cst or varying with z0 are very similar (11.5 and 10.96 cm). Is there a significative improvement with Cd=f(z0,H)Âǎ? It is not justified why the authors chose Cd=f(z0) rather than Cd=cst for the following. Indeed, SET4, could have been made with Cd varying spatially, instead of z0 varying spatially.**

We agree with the reviewer that a decrease of the cumulative errors of 0.54 cm between simulations with CD=cst and simulations with CD=f(z0,H) is small. To examine into details the question of significance of this result, we plotted the maps of relative differences (in both amplitude and phase) for all four waves between the two simulations (CD=0.9 10-3 m and CD=f(z0=1.5 10-5, H) taking FES2014b-with-assimilation as a reference (Fig. 2 below) i.e:

difference= abs(A-C)-abs(B-C)/abs(A-C)

where A corresponds to the simulation SET2 with CD=f(z0=1.5e-5, H), B corresponds to the simulation SET1 with CD=0.9 10-3 m and C corresponds to the product FES2014b-with-assimilation.

The results are heterogeneous and vary from a wave to another. For K1, the positive values almost all over the basin indicate that the simulation with a constant CD has the smallest difference to the reference FES2014b. However, the solution is improved only by ∼1% compared to simulation with CD=f(z0) (Fig.2). Difference values for O1, M2 and S2 are mostly negative. This means that simulation with CD=f(z0) slightly improves (from 0.5% to 5%) the solution compared to simulation with a constant CD.

These results show that differences between both simulations relatively small and not significant (smaller than 5%), neither in average nor locally : they thus finally suggest that the sensitivity of the tidal solutions over the GoT to the choice of bottom friction paramaterization (constant CD vs. spatially varying with CD=f(z0) is limited compared

Interactive
comment

to the sensitivity of the solutions to bathymetric changes.

Figure 2: Relative differences (in %) between simulation with CD=f(z0=1.5e-5,H) and simulation with CD=0.9e-3 m compared to FES2014b-with-assimilation for the tidal harmonics of O1, K1, M2 and S2.

Furthermore, we remind that we initially tested these two parametrizations (CD=cst and CD=f(z0,H)) to assess the dependence of CD to bathymetric changes over the plateau. The similar cumulative errors (11.5 and 10.96 cm) therefore confirmed the low dependence of CD to bathymetric changes in the GoT shallow waters, and suggested that either of the two CD formulations could be used for 2D shallow water tide modeling in the region. We also tested the parametrization with a CD=f(z0,H) for 3D modeling purposes (as it is the end goal of our study). Indeed, in 3D modeling, the formulation of CD depends on the z0 and on the reference height in the logarithmic layer above the bottom zb as follows:

CD=(k/ln(zb/z0))ˆ2

where k=0.41 is the Von Karman constant. The use of such parametrization ensures that the bottom drag estimation in the model is independent of the reference height in the logarithmic layer above the bottom, which is critical in 3D models with variable vertical grids, motivating our choice to spatially vary z0rather than CD.

–> Following this comment, we have included a new figure (as Fig. 11) in our revised version of the manuscript, which synthesizes the difference of performance of our simulations. The difference concerning the use of a constant CD vs. a varying CD =f(z0) is discussed consistently with our answer above in lines 623-636.

**27-A lines 521: 'Spatially varying uniform friction parameters induce the best results on the tidal solutions rather than on uniform parameters.' Is the improvement significative?**

–>Following our answer to #29 – C just above regarding the significance of this improvement, the sentence "Spatially varying uniform friction parameters induce the best results on the tidal solutions rather than on uniform parameters." has been changed to "Spatially varying uniform friction parameters only slightly improve the tidal solutions compared to uniform parameters." lines 553-554 of the revised manuscript.

Part 2- Moreover, the fact of varying spatially the bottom friction (here z0) is not very convincing. Indeed, the choice of three areas (SET4) finally leads to results very close to one area (SET2) as two of the three areas have the same z0 than the SET2 optimized (z0=1.5 10-5m) and without surprise, the cumulative errors are also very similar, even if slightly better with SET3 (10.43cm instead of 10.96cm for SET2). More disturbing is the fact that increasing the number of areas, the results are worse (SET5 with a cumulative error of 12.29cm instead of 10.43cm for SET4). This raises the question of the the robustness of the method.

This general comment regards the question of the choice of a spatially variable z0. Since this question is also addressed in #27-B, #30 and #38, we regroup here the answers to those related comments to avoid repetitions make our answer clearer.

**30-A - line 585 and followingĂă: 'Sensitivity to the value of spatially varying roughness length (SET3, SET4, SET5). This part could be separate in two parts 'Sensitivity to a quadratic or linear stress' which correspond to SET3 compared to SET2) and 'Sensitivity to a roughness length varying spatially' (which corresponds to SET4 and SET5 compared to SET2).**

–>Following this comment, we have divided the section into two parts, line 642 and line 698.

**30-C. SET4Ăă: the three areas are finally only two areas (Regions 1 and 2 with the same z0), and the z0 in areas 1 and 2 is the one corresponding to SET2 optimized (z0=1.5 10-5 m). Finally, SET4 is not so different from SET2, with very similar cumulative error (10.43 cm instead of 10.96 cm). Is the improvement significative?**

**38-line 791: 'the regionalisation of the roughness length into three regions, for addressing the issue of representing the complexity of seabed composition and morphology, moderately improved the accuracy of our simulation, with a lowest cumulative error for all four waves of 10.43 cm', instead of 10.96 cm. We totally agree with this conclusion, is the improvement significative enough?**

We agree with the reviewer that simulation named TEST 6 (from SET4) is finally similar to the simulation using CD=f(z0=1.5e-5, H) (from SET2) and that therefore, TEST6 and CD=f(z0=1.5e-5, H) show similar cumulative errors. To assess the significance of this result, we plotted the relative difference of the tidal harmonics for the four waves between simulation TEST6 and simulation using CD=f(z0=1.5e-5, H), taking FES2014b-with-assimilation as a reference(Fig. 3) i.e.:

difference= abs(A-C)-abs(B-C)/abs(A-C)

where A corresponds to the simulation SET3 from TEST6, B corresponds to the simulation SET2 with CD=f(z0) and C corresponds to the product FES2014b-with-assimilation (reference).

Similar to the previous results presented in General comment #3 part 1, relative differences between the simulations are relatively small, heterogeneous over the GoT basin and varying from a wave to another. For O1, the positive values all over the GoT indicate that simulation with CD=f(z0) (SET2) produces smaller differences to the reference (Fig. 3). This improvement compared to simulation " TEST6" is however of only ∼2%. On the opposite for K1, simulation " TEST6" (SET3) produces smaller differences to the reference mostly all over the basin. The improvement is however smaller than ∼3%. Regarding the semi-diurnal waves, simulation " TEST6" shows smaller differences to the reference along the northern coasts of the GoT and along a branch near the southern open boundary of the domain, while simulation with CD=f(z0) shows the smallest difference in the rest of the basin. Once again, the improvements from a simulation to another are relatively small (∼1%).

From these results, we can conclude that our simulation using a spatially varying z0 only slightly improves the solution of K1, does not improve the solution of O1, and locally improves the solutions of the semi-diurnal waves compared to simulation with CD=f(z0). Given the very small differences between the two simulations (∼3%), we can conclude that the improvement from a simulation to another is not significant.

These results therefore suggest that tidal solutions in the GoT are more sensitive to bathymetric changes (as discussed in the answer to General Comment 4 below) rather than to changes in bottom friction parameterization: i.e. changes in seabed composition and morphology.

–>Following this comment we have included a new figure (as Fig. 15) in our revised version of the manuscript, which synthesizes the difference of performance of our simulations and added comment accordingly to the above discussion, lines 679-688.

Figure 3: Relative differences (in %) between simulation with CD=f(z0=1.5e-5,H) and simulation from TEST 6 compared to FES2014b-with-assimilation for the tidal harmonics of O1, K1, M2 and S2.

From these results, we can conclude that our simulation using a spatially varying z0 slightly improves the solution of K1, does not improve the solution of O1, and locally improves the solutions of the semi-diurnal waves compared to simulation with CD=f(z0). Given the relatively small differences between the two simulations (∼3%), the improvement from a simulation to another is not significant. These results furthermore suggest that tidal solutions in the GoT are more sensitive to bathymetric changes rather than to changes in bottom friction parameterization: i.e. changes in seabed composition and morphology.

**27-B. Moreover, 12 areas give worse results than three areas, and three areas correspond finally to only two. Is the method robust enough?**

**30-D. SET5Âǎ: how to explain that it is worse with 12 areasÂǎ? Why don't we converge also to z0=1.5 10-5 mÂǎ? It could be interesting to include these results and analyse them, otherwise, we don't have enough elements to understand, and we can wonder if the method is robust enough.**

**27-B. Moreover, 12 areas give worse results than three areas, and three areas correspond finally to only two. Is the method robust enough?**

We thank the reviewer for highlighting this issue which questions the robustness of our method, and pointing out a weakness of our previous tests, that did not cover the whole range of values. We ran our simulations for SET5 again with 12 areas, covering the whole range of z0 values. We obtain the smallest error when affecting a value of z0=1.0 10-4 m to areas n°5 to 12, and a value of 1.5 10-5 m to areas n°1 to 4: this configuration corresponds exactly to TEST6 from SET4. We thus finally found the same cumulative error of 10.43 cm in SET4 and SET5. This therefore confirms the robustness of our method. This error is now corrected in the text lines 692-693.

Part 3-The last point is that choosing for the mud area a linear expression instead of a quadratic one has no significative influence (SET3).

Since the significance of the choice of linear vs. quadratic expression for the mud area addressed in this General comment #3–part 3 is also addressed in #19, #28, #30-B and #39, we regroup here the answers to those related comments to make our answer clearer.

**19- line 332: 'The quadratic parameterization maybe obsolete and a linear parameterization more adequate'. Is there a justification or a reference for this sentence? The results will show the contrary.**

**28- line 522: 'However, prescribing a linear parametrization in supposed fluid mud areas does not allow to significantly improve the solutions' How to explain this?**

**30-B. SET3 has been conducted with a constant value of r and varying value of z0. Without surprise, optimization leads to z0=1.5 10-5 m, which is the optimized value of**

SET2. Why not fixing the z0 value (z0=1.5 10-5 m) and make vary the r valueĂă? This could lead to an optimized r value, probably different from 1.18 10-4 m from Le Bars et al. (2010), and perhaps results would show more sensitivity to a quadratic or linear stress (depending on r value).

line 794: 'Finer local adjustments of the roughness length or the choice of a linear velocity profile in the area of fine mud, did not improve the accuracy of our simulations.' We totally agree with this conclusion, how to explain that there is no improvement?

**39- line 794: "Finer local adjustments of the roughness length or the choice of a linear velocity profile in the area of fine mud, did not improve the accuracy of our simulations." We totally agree with this conclusion, how to explain that there is no improvement?**

The reference is 'Le Bars et al. 2010'. We understand the reviewer's concern on this particular point. Our hypothesis is that there is no improvement in simulations from SET3 (linear velocity profile in the area of fine mud) because the model sensitivity to the bottom friction is limited in the area since tidal energy fluxes and dissipation are themselves limited in this particular area. To confirm this hypothesis, we plotted maps of tidal energy fluxes (Fig. 4) and maps of bottom dissipation (Fig. 5) for all four waves. It appears that a branch of intense tidal energy fluxes enters the GoT from the south-eastern boundary (south of Hainan island) for O1, K1 and M2 (Fig. 4), and is dissipated along the western coasts of the island (Fig. 5). Energy fluxes also pick up in the Hainan Strait due to combined entering and exiting fluxes and are dissipated along this same strait. However, along the Red River Delta, both tidal energy fluxes and bottom dissipation are extremely limited for all four tidal constituents. Therefore, changes of dissipation mode along the Red River Delta, i.e. in the area of fine mud of SET3, do not significantly affect the tidal solutions as tidal energy flux and bottom dissipation are very weak. –> Following this comment, Le Bars et al. 2010 has been added line 353-354, explanations on why simulations from SET3 were not improved with the use of a linear velocity profile in the area of fine mud are now added lines 657-666 and Fig. 5 (as Fig. 14 in the manuscript) is added to the revised manuscript,

and this conclusion is acknowledged in our revised paper, line 927-928.

Furthermore, we could have played with the value of r, which was previously optimized by Le Bars et al. (2010). However, it would not have significantly changed our results since we have seen that the choice of a linear parameterization does not significantly improve the tidal solutions. –>This suggestion is now acknowledged in the text lines 662-666.

Figure 4: Tidal energy flux (W m-1) for O1, K1, M2 and S2 computed from model outputs of simulation 'TEST 6'.

Figure 5: Bottom dissipation flux (W m-2) for O1, K1, M2 and S2 computed from model outputs of simulation "TEST 6".

Part 4-Finally, the sensitivity study allows to optimize bottom parameters (e.g. r, Cd or z0), but the sensitivity to a parameterization rather than another is not clearly justified in the paper (except a cumulative error slightly lower). A discussion would be welcome to clarify this point: How significant is the choice of parameterization between Cd=cst or Cd=f(z0) or . . . ? Why is it here not significant ?

We agree with the reviewer that the choice of a simulation rather than another was not clear in our manuscript. Our answers made above to General comment #3 - part 1 and 2 and specific comments #27 to #30 provide detailed elements to answer to this general issue of the significance of our simulations. Finally, our results first allowed to optimize the values of the coefficients associated with the different parameterizations, second showed that the choice of the bottom friction parameterization does not significantly affect the performance of our simulations and that a simple constant Cd parameterization can be used. This will be useful for the implementation of the 3D structured model that will be used for the study of the sedimentary dynamics in the region. Third, our results regarding the influence of the bathymetry dataset show that this bathymetry is finally the key ingredient for tidal representation in this specific structured grid configuration. –>This represents the main message of our paper, and was

highlighted in the conclusion of our revised paper, lines 930-934.

General comment #4 Bathymetry sensitivity Part 1- Results show a clear improvement between GEBCO and improved bathymetry. However, the use of bathymetry from nautical charts could reduce the depth (because charts are made for navigation purpose, see explanation in the specific comments) and lead to an overestimation of the tide. The improvement between the results and existing atlas FES2014b-hydro (without assimilation) is not so significative. For example, if we look at M2 (Figure 6), results show lower errors in the South but greater error in the North, with significant differences between TKN model and FES2014b-hydrodynamics. The overestimation of the amplitude nearshore could be partly due to the use of the bathymetry from the nautical charts.

This general comment regards the use of nautical charts. Since this question is also addressed in specific comment #14, we regroup here the answers to those related comments to avoid repetitions and to make our answer clearer.

**14- line 261 ă: 'digitalized nautical charts' Charts are made for navigation, and near the coast, the hydrographer generally chooses the lower soundings (shallow waters) for security purpose. As a consequence, the bathymetry from nautical charts in coastal areas gives shallower waters than ń ăreal ă ż bathymetry. This should be mentionned, as it could partially explain the overestimation of tidal amplitude with TKN compared with FES2014b-hydrodynamics.**

We thank the reviewer and agree with his/her comment that bathymetry from nautical charts could lead to an overestimation of the tidal amplitude, and that differences between simulated amplitudes for M2 and S2 could be partly explained by bathymetric underestimations. –>We now mention in the text the potential issues induced by the use of nautical charts on the bathymetry dataset, lines 279-280, and discuss this issue in the manuscript lines 810-812 and lines 829-831.

Part 2-Note that the part of the FES2014b outside the Gulf of Tonkin could be masked, as it is not modelled in TKN. Otherwise the scale of the figure is not appropriate, e.g.

in Figure 7 for S2 the scale ranges from 0 to 0.33 m (note that there is no unit on the figures) whereas the maximum in the Gulf of Tonkin is only 0.19 m.

We understand the reviewer's concern. The issue of the color scale however only occurs for one of the 4 waves (S2), and moreover we do think it is valuable for the manuscript to keep unmasked the tidal amplitudes outside the GoT as it helps the reader understanding the amplitude patterns inside the GoT. –>We therefore decided to keep the values of FES2014b outside unmasked.

Part 3- Another point is that comparisons were made only between the model and altimetry, and comparison with tide gauges would be of great interest for the paper. For example, on Figure 6 authors could add dots at tide gauges locations colored with corresponding M2 amplitude.

This general comment regards the use of tide gauges data. Since this question is also addressed in #41, we regroup here the answers to those related comments to avoid repetitions make our answer clearer.

**41- line 806: 'Our resulting configuration brought a clear improvement in the tidal solution compared to previous 3D simulations from the literature and to the tidal atlas FES2014b (without data assimilation) for the semi-diurnal waves.' The improvement compared to FES2014b is not so clear if we look at Figure 6 for example. The addition of tide gauges data should greatly help to qualify the results.**

We agree with the reviewer's suggestion that comparisons of simulated tidal solutions with tide gauges is of great interest for this study. Therefore, we compared the tidal harmonics of TKN, TKN-gebco, FES2014b-without-assimilation and FES2014b-with-assimilation with tidal harmonics from 11 gauge stations along the Gulf coast (provided by the International Hydrographic Organization, IHO). The positions of these stations are now shown in Fig. 5 a of the manuscript (and on Fig. 6 of the present text).

Figure 6: O1 tidal amplitude (in m) from FES2014b-with-assimilation superimposed

[Figure]

with locations of tide gauges (black cross). This figure now corresponds to Fig. 5 a of the revised manuscript

RMS* errors between modelled and observed tidal harmonics from tide gauges are now shown in Fig. 16 b of the manuscript (and on Fig. 7 of this document). First, compared to TKN-gebco, TKN gives smaller errors for all four waves considered: RMS* for K1 are reduced by ∼40% in TKN compared to TKN-gebco and RMS* for M2 are reduced by ∼45% in TKN compared to TKN-gebco. This result again confirms that TKN configuration improves the tidal representation over the GoT. Second, TKN simulation shows also smaller RMS* errors than FES2014b-without-assimilation simulation for O1, M2 and S2. In addition, TKN even minimizes S2 RMS* errors compared to FES2014b-with-assimilation. Our improved configuration however fails to improve the solution of K1, compared to the two FES2014b products. This last result is due to the fact that the unstructured grid used in FES2014b-(with and without-assimilation) is better adapted for the representation of the complex coastal topography that the structured grid that will finally be used in our tridimensional model and that is therefore used in TKN configuration. –>These tide gauges data are presented in the revised manuscript lines 537-542, and the comparison of our simulations with those tide gauges data is now discussed in section 3.2.1 (average assessment over the domain) in the manuscript, lines 737-743.

Figure 7: RMS* errors between numerical simulations (TKN, TKN-gebco, FES2014b-without-assimilation, FES2014b-with-assimilation) and tide gauges harmonics for O1, K1, M2 and S2.

Part 4-Finally, the TKN model has clearly errors greater than FES2014b-synthesis (with assimilation) which shows that the first objective (improve tidal model) is not really reached.

We agree with the reviewer that our optimized simulation did not improve the tidal solution compared to FES2014b-synthesis (now called FES2014b-with-assimimilation,

following the reviewer's Specific comment #21). However, as explained above in the answer to General comment #1, our primary goal is to improve the tidal solutions in a model without data assimilation and based on a finite difference grid with the objective of implementing a structured 3D model, which is reached. –> This question regarding the central objective of this work is better detailed in the introduction of our revised manuscript, lines 183-192.

General comment #5 Maybe the paper would be clearer if the part 3.2 'Sensitivity to the bathymetry and assessment of tidal solution' could be separated in two parts, as here sensitivity to the bathymetry and assessment of tidal solutions are mixed, which is quite confusing. Results of the paper could be reorganized into three partsÂă: 1) Sensitivity to bottom stress parameterization, that would lead to parameterization Cd=f(z0) with z0 varying spatially 2) Sensitivity to the bathymetry, that would lead to TKN bathymetry 3Âă) Assessment of tidal solutions, that would show tidal improvement compared with Minh et al. (2014) and Chen et al. (2009), but not so clear improvement with FES2014b-hydrodynamics (e.g. Fig. 6) and clearly, no improvement compared with FES2014b-synthesis (with data assimilation). This underline the importance of data assimilation, and the need to go on developing satellite mission and in-situ campaigns, despite the great improvements of numerical models in the last decades.

We thank the reviewer for suggesting an outline that would improve the clarity of the 'Results' section of the manuscript. –>The last is acknowledged in the conclusion of our revised manuscript lines 947-949 and the recommendations have been taken into and this section is now organized as followsÂă:

3. Results 3.1 Model sensitivity to bottom stress parameterization 3.1.1 Sensitivity to a constant or varying CD (SET1 and SET2) 3.1.2 Sensitivity to the value of spatially varying roughness length (SET3, SET4, SET5) 3.1.2.1 Sensitivity to a quadratic or linear stress 3.1.2.2 Sensitivity to a spatially varying roughness length 3.2 Sensitivity to the bathymetry 3.2.1 Average assessment over the domain 3.2.2 Spatial assessment of tidal solutions 3.3 Assessment of tidal solutions with previous studies

II Specific comments

**1- line 45: the "strong improvement (compared to pre-existing tidal atlases)" of tidal solution is not so clear compared to FES2014b-hydro (without assimilation, see general comment for Figure 6). Moreover, the model errors are really greater than FES2014b-synthesis (not surprising, as this last one is with data assimilation.**

The same issue is raised in #32 and 36, so we answer together to those comments.

**32 - It is not so clear why TKN and TKN-gebco show bigger errors than FES2014b-hydrodynamics. We understand then K1 is less sensitive to bathymetric variations, but this is probably not the only explanation.**

**36- line 763: it is clear that the new bathymetry improves the results compared with GEBCO, but it is not clear if the final configuration TKN is improved compared with FES2014b.**

We completed Table 2 of the revised manuscript, with the MAE of amplitudes and phase of M2, S2, O1, K1 constituents between the two FES2014b products and satellite altimetry (following the suggestions of specific comments #43) to clarify this point. Moreover, we added to Fig. 16 (of the manuscript) which provides a comparison with tide gauges, following General comment #4-part 3 and specific comment #41. Both Table 2 and Fig. 16 (of the manuscript) show that TKN gives the smallest errors in amplitude and phase to satellite altimetry of all four waves, compared to FES2014b-without-assimilation (except for the phase of S2), for the semi-diurnal waves when comparing cumulative errors to altimetry and for O1, M2 and S2 when comparing cumulative errors to tide gauge data (please refer to our response to Specific comment #41 for more details). This confirms that TKN simulation improves, though slightly, the tidal results compared to FES2014b-without-assimilation. However, TKN does not improve the MAE compared to TKN-with-assimilation (except for the amplitude of S2).

–>This is now detailed in the text lines 737-767, and in the abstract, the sentence

'strong improvement (compared to pre-existing tidal atlases)' is now changed to 'slight improvement (compared to pre-existing tidal atlas without data assimilation)' line 45.

Finally, another reason to the bigger errors observed for the diurnal waves with TKN and TKN-gebco compared to FES2014b-hydrodynamics (now called FES2014b-without assimilation) is due to the fact that FES model is specifically dedicated to tidal simulation in coastal areas. Indeed, the unstructured triangle grid meshes of FES model allow both high flexibility and resolution over the complex topography of coastal areas, while regular quadrangle structured C-CGRID meshes fail to represent. –>This is now discussed in the revised manuscript lines 732-735 and in the conclusion lines 943 946.

2- line 101 and followingÂǎ: add in Figure 1 geographic elements quoted in the text as for example Gulf of Tonkin, Hainan Strait, Hainan Island, Zhanjuang Peninsula, Qinjjiang, Nanliu, Yingzai rivers, Hai Phong harbour . . .

–>We thank the reviewer for this suggestion, these elements have been added to Figure 1 of the manuscript.

3- line 153 and followingÂǎ: precise the expression of the tidal factor, and the existence of the four regimes.

–>Following the reviewer's suggestions, the expression of the tidal factor $F=(O1+K1)/(M2+K2)$ as well as the definition of the four regimes have been added to the text lines 157-158.

4- line 153-162Âǎ: maybe a map of F would be useful, as the variation of F is here described spatially.

As suggested by the reviewer, we plotted a map of F where we clearly see that the tidal regime over the GoT is mainly diurnal (F>3), except in the southwesternmost part of the GoT where the regime is mixed dominantly diurnal (1.5<F<3) (Fig. 8). –> this map is now included in our manuscript (as Fig. 2), line 163.

Figure 8: Map of tidal form factor F computed with the amplitudes of tidal waves obtained from FES2014b.

5 –>now in response to General comment #1

6- line 189: 'in poorly sampled regions', precise in terms of what, bathymetryĂă? Sea levelĂă? Tidal currentsĂă? Tide gaugesĂă?

–>Following this comment, details have been given lines 197.

7- line 193Ăă: 'in situ data and soundings are consequently rare and extremely valuable', is it possible to list these data in the areaĂă? Particularly tide gauges may be of great interest for this study.

–> Following the reviewer's suggestion, we added to the text the list of tide gauges available in the area lines 537-542. These tide gauge data are now used to answer General comment #4 and Specific comments #41.

8- line 230Ăă: why did you choose Bing as 'the reference'? Is there a paper reference for thisĂă?

In this study, Bing has been chosen for the accessibility to its opendata. As one of the goal of this study was to propose a simple method to improve tidal solutions, which could be easily adapted to other study areas, we chose to use opendata maps accessible by anyone. To our knowledge, no reference paper exists about this. –> Following this comment we have justified our choice of Bing in the revised version of the manuscript, line 239.

9 and 10 - line 235: 'OpenStreetMap shoreline is most of the time shifted', is there an explanation for thatĂă? - line 236: 'The GSHHG dataset suffers from the same problem but shifted by up to 500m eastwards.', this shift is here very significant, is there an explanation for that?

We regroup the reviewer's comment #9 and #10 as they both address a similar issue

and we understand the shifts observed in both shorelines datasets can arouse curiosity. We could not find any available information concerning this shift. We believe OpenStreetMap and GSHHG datasets over our area of study were both constructed with composites data, as for example from digitalized nautical charts and local topography maps, which could be shifted if collected before accurate GPS measurements in the area. –>We added a sentence to acknowledge this issue in the revised version, lines 246-249.

11- line 238: 'matching the reality', what is the reality, and why? Bing maps?

In this sentence indeed, the term 'matching the reality' corresponds to Bing maps. –>This is now detailed in the text line 250.

12- line 240: shorelines from POCViP, is there a reference paper for this software? What are the data behind this software?

In this study, POCViP software was used to construct the GoT shorelines and Red River waterways. This software allows the user to build or 'draw' polygones (with segments and nodes) that can be saved as shapefiles. In our case, the polygones were built following the Bing maps that were previously georeferenced with POCViP. This software does not provide any shoreline data by itself. There is no reference paper for this software but information can be found here: https://mycore.corecloud.net/index.php/s/ysqfIlcX5njfAYD/download –>this information is provided in the paper, line 256..

13- line 257: the GEBCO resolution is of approximately 1 km. There is an important difference between the grid resolution and the data resolution, as the data resolution could be lower than the grid resolution (and interpolated). Of which resolution are we talking here? Grid resolution? Do we know what is the data resolution?

The GEBCO dataset used in this study is provided on a 30 arc-second interval grid, but is largely based on a database of ship-track soundings, whose resolution varies

from a region to another and can hence be locally finer than 1 km, as explained on the GEBCO website (https://www.gebco.net/data_and_products/historical_data_sets/ ). –> this was mentioned in our manuscript, lines 268-272.

14 –> now in response to General comment #4

15- line 285: 'TONKIN-bathymetry dataset is not considered as the truth', rather say that due to sampling, there are still uncertainties on the bathymetry. Also mention problems linked with nautical charts (shallower water, see comment above).

–> In link with the previous comment #14, this suggestion is added line 304-306.

16- line 300 and following, no reference of TUGO for 'storm surge simulations'Âǎ?

–> Indeed, 'Storm surge' has been removed line 312.

17- line 319: version of the code is a little bit complicated. . . ' 2616:78a276dd7882 of 2018-07-22 13:17 +0200'

We have changed the name of this version to "version 4.1 2616". –> This has been changed in the text line 338.

18- line 320: TKN is the name of the code or of the configuration? Not clear here.

TKN refers to the name of the configuration. –>This is now detailed line 339.

19 –> now in response to General comment #3–part 3

20- line 340: the 'final goal' i.e. hydrodynamic-sediment transport with SYMPHONIE-MUSTANG could be introduced earlier. –>In link with specific comment #5, the final goal of modelling hydrodynamic-sediment transport has been introduced lines 183-192.

21- line 337: the names FES2014b-hydrodynamics and FES2014b-synthesis are not really explicit. Choose for example FES2014b-without-assimilation and FES2014b-with-assimilation or something else but more explicit. –> FES2014b-synthesis has

been changed into FES2014b-with-assimilation and FES2014b-hydrodynamics into FES2014b-without-assimilation throughout the manuscript and figures.

22- line 383: is there a reference for FES2014bÂă?

–> Yes, Carrère et al. (2016), this reference for FES2014b has been added line 384.

23- line 402 and following:2.3.1.1 Bottom stress parameterization, this section is not clear enough. The three parameterization (2) (3) (4) are finally two parameterizations (1a) and (1b), the first one with Cd=constant or Cd=f(z0,H) and the second one with r=constant. Particularly, the sentence 421 is not clear 'In this study, we test three commonly used parameterizations: a constant drag coefficient Cd assuming a constant speed profile or a linear speed profile, and a drag coefficient Cd depending upon the roughness height z0'. This paragraph could be rewritten to be clearer.

We thank the reviewer for pointing out this unclear explanation. –> Following this comment we have re organized the text and the order of equations accordingly from lines 427-483. We hope it will improved the clarity of this section.

24- line 447: 'In presence of fluid mud,...Âă' repetition, yet said before, line 415.

–>Thank you for spotting this repetition, the sentence has been changed now line 471.

25- line 457: (2) and (4) are linked with (1a) parameterization whereas (3) is linked with (1b) parameterization. The way the parameterization are presented is confusing.

–> We agree and we made the changes accordingly (see response to Specific comment #23).

26- line 464: 'two of the parameterizations decribed aboveÂă: a quadratic bottom stress with a uniform drag coeffiencient Cd (Eqs. 1a and 2) and a logarithmic variation of Cd depending on a uniform bottom roughness height z0 (Eq. 4)' is not very clear. It would be more appropriate to talk about a quadratic bottom stress with a drag coefficient constant (Cd=cst) or varying with the roughness length (Cd=f(z0,H)).

We thank the reviewer for this suggestion that will make the manuscript clearer. –>We have changed the text accordingly from lines 486-494 and throughout the results section.

27 to 30 –> as explained at the beginning of this document, those comments where gathered to General comment #3 above to make our response clearer and more consistent.

31- line 622: '3.2 Sensitivity to the bathymetry and assessment of tidal solution', this section could be split in two parts, see comment in general comments.

–>as explained in our response to General comment #5 (see above), we have reorganized section 3 accordingly.

32 –> see answer to specific comment #1

33- line 686 and following: TKN is better than Minh et al. (2014) and Chen et al. (2009). Why? This could be explained.

We believe the improvements of our tidal solutions compared to the solutions of Minh et al. (2014) and Chen et al. (2009) are due to two main reasons: - First, to the model efficiency in tidal modelling. Indeed, the model used in our study, T-UGOm, is a state-of-the-art tidal model and is specifically developed for tidal modeling purpose. Therefore, it is expected that simulations from T-UGOm would show better results than 3D simulations performed with ROMS_AGRIF, as used by Minh et al. (2014) and with ECOM as used by Chen et al. (2009), which both are hydrodynamical models and not specifically conceived for tidal modelling. - Second, our model configuration has been optimized for tidal modeling purpose, in terms of grid resolution, bathymetry accuracy and resolution and bottom friction parametrization, whereas the model configuration of these two previous studies may not have been specifically optimized for tidal modelling purpose. –>an explanation has been added in the revised version of our manuscript, lines 860-866.

[Figure]

34- line 686 and following: the acronym SLA is not detailed, are we talking about Sea Level Anomaly? This term is generally corrected from tide. It is not clear here.

We thank the reviewer for spotting this mistake and understand it can generate confusion for the reader. We compared our results to altimetry-derived ocean tide harmonic constants and not to sea level anomaly. –>We now specify this line 713-714 and we determined an other acronym "AH" (for Altimetric Harmonic), that is used in the following lines, which we hope will be less confusing for the reader.

35 –> now in response to General comment #1

36 –>now in response to Specific comment #1

37- line 784: 'the use of a constant Cd parameterization or the use of a Cd depending on the roughness length led to fairly similar results'. We totally agree with this conclusion, that could appear earlier in the paper (in the results).

–> following this comment, we included this conclusion earlier, in the Results part, lines 635-636.

As a consequence, why choose a Cd depending on the roughness length instead of a constant Cd for SET3/SET4/SET5?

We chose to vary the roughness length to take into account and represent, the seabed's morphology and composition. As the end, the values of CD when depending on z0 are very similar to 0.9 10-3 m (see Fig. 1 in General comment #2), taking a constant or a varying CD would have led to similar results.

38- see answer to General comment 3 part 2 above.

39- see answer to General comment 3 part 3 above.

40- line 800: 'results therefore quantitatively showed the importance of the bathymetry and shoreline dataset and of the choice of bottom friction parameters for the representation of tidal simulations over a shallow area like the GoT'. This could be clarified.

The results show that the choice of the bottom stress parameterization is not so important (e.g. SET2 optimized with Cd=cst gives similar results than SET3 optimized with Cd=f(z0,H), in terms of cumulative errors), but the value of the bottom parameter (e.g. Cd=0.9 10-3 m for SET1 or z0=1.5 10-5 m for SET2) is important, as it impacts clearly the cumulative error (e.g. Figure 8).

We agree with the reviewer that this sentence should be clarified : the main point is that the choice of the parameterization does not significantly affect the performance of the model, but that the choice of the value of the friction parameter, for a given parameterization, is important. –> following this comment, we modified the text in our manuscript (lines 930-934).

41 –> now is addressed in our response to General comment #4

42- line 813: 'Using bathymetry data available from digitalized navigation charts was a relatively simple way (compared to performing additional in-situ measurements) to significantly improve the representation of topography in the coastal and estuarine areas of the GoT.' We agree, however, as mentioned above, bathymetry could be underestimated (shallower waters) because charts are made for navigation and the shorter soundings are choosen for security reasons. The use of nautical charts could then lead to an overestimation of the tidal amplitude in some coastal areas.

We agree with the reviewer's comment, see our responses to specific comment #14. –> The issue of the use of nautical charts is now mentioned in the Conclusions section, lines 954-956.

43- Table 2, FES2014b hydrodynamics and FES2014b-synthesis are clearly missing

–> We took into consideration this comment and added the values of MAE of amplitude and phase of O1, K1, M2, S2 between FES2014b-with-assimilation and FES2014b-without-assimilation with satellite altimetry in Table 2, and we discussed these results lines 737-743.

[Figure]

Please also note the supplement to this comment:
https://www.geosci-model-dev-discuss.net/gmd-2019-40/gmd-2019-40-AC2-supplement.pdf

————————————————

[Figure]

Fig. 1. $C_D = f(z_0, H)$ with $z_0 = 1.5e^{-5}$ m

[Figure]

**Fig. 2.**

[Figure]

Fig. 3.

**Fig. 4.**

[Figure]

**Fig. 5.**

**Fig. 6.**

[Figure]

**Fig. 7.**

[Figure]

**tidal form factor F**

**Fig. 8.**

---

## Author Response (AR2)

The authors have clearly improved the manuscript and made it really clearer. It is suitable for publication, with minor revisions :

**We thank the reviewers for his/her careful read of our manuscript and for thinking our manuscript is suitable for publication after minor revisions. Please see our responses below and our changes in the manuscript (in red).**

1) Line 380 : M4 is a non linear interaction tidal component, please replace « all linear interactions » by « all non linear interactions ».

**➔ This is now changed in the manuscript line 382.**

2) Fig. 2 : the legend could specify the tidal regimes « Map of tidal form factor F computed with the amplitudes of tidal waves O1, K1, M2 and S2 obtained from FES2014b-with-assimilation. 0<F<0.25 corresponds to semi-diurnal regime, 0.25<F<1.5 mixed primarily semi-diurnal, 1.5<F<3 mixed primarily diurnal and F>3 diurnal regime ».

**➔ The legend of Fig. 2 is now changed accordingly.**

3) Comme on Fig. 5-8 a : it would have been more valuable to plot a coloured for each tide gauge, the colour corresponding to the amplitude of the harmonic component (see previous comment « on Figure 6 authors could add dots at tide gauges locations coloured with corresponding M2 amplitudes »).

**We apologize for missing this comment in the previous reviewer's comments.**
**➔ Following his/her suggestion, we have added coloured circles to Figs. 5-8 a ; the colour corresponding to the amplitude of the tidal harmonic measured at each gauge station.**

---

## Author Response (AR3)

Dear Violaine,

Thanks for addressing the reviewer's comments. The only remaining issue, which I should have picked up last time, is that the Zenodo citations in the code and data availability section are presented incorrectly. You have included URLs for locations on the Zenodo website. This is incorrect for two reasons. First, you should be using the DOI, not the URL, and second the citations should be presented in the bibliography and cited from the text like any other reference.

If you navigate to any of your Zenodo pages, e.g. https://doi.org/10.5281/zenodo.2669396, and look to the bottom right, it shows you the correct form of the bibliography entry, in that case it is:

Piton Violaine. (2019). Grid for T-UGO [Data set]. Zenodo. http://doi.org/10.5281/zenodo.2640763

An example of a GMDD manuscript with correct Zenodo citations in the code and data availability section is: https://www.geosci-model-dev-discuss.net/gmd-2019-86/gmd-2019-86.pdf

Regards,

David

**Dear David,**

**I changed the Zenodo citations in the Code and/or data availability section, as well as in the References, according to your suggestions. I hope this is correct now.**

**Best,**

**Violaine**